# FOCUSDIFF: ADVANCING FINE-GRAINED TEXT-IMAGE ALIGNMENT FOR AUTOREGRESSIVE VISUAL GENERATION THROUGH RL

## ABSTRACT

Recent studies extend the autoregression paradigm to text-to-image generation, achieving performance comparable to diffusion models. However, our new Pair-Comp benchmark – featuring test cases of paired prompts with similar syntax but different fine-grained semantics – reveals that existing models struggle with fine-grained text-image alignment, thus failing to realize precise control over visual tokens. To address this, we propose FocusDiff, which enhances fine-grained text-image semantic alignment by focusing on subtle differences between similar text-image pairs. We construct a new dataset of paired texts and images with similar overall expressions but distinct local semantics, further introducing an improved GRPO-based algorithm to emphasize such fine-grained semantic differences for desired image generation. Our approach achieves superior performance on existing text-to-image benchmarks than many industry-leading models and also outperforms most prior prominent methods on PairComp. Anonymous Project: this link.

## 1 INTRODUCTION

Witnessing the scalability of autoregression (AR) in large language models (LLMs OpenAI, 2023), recent studies have extended the AR paradigm to text-to-image generation, achieving performance comparable to diffusion models (Labs, 2024). Some work (Wang et al., 2024; Chen et al., 2025b) encodes images into discrete tokens and transforms image generation into a next-token-prediction task. Additionally, some studies (Deng et al., 2025; Wu et al., 2025b) further explore hybrid approaches that integrate AR with diffusion to harness the strengths of both.

Despite extensive vision-language alignment, existing models that incorporate the AR paradigm still struggle with precise control over images based on text conditions. To elucidate this problem, we first introduce the **PairComp benchmark**. Unlike typical text-to-image benchmarks (Ghosh et al., 2023) with a single prompt per test case, each case in PairComp consists of two prompts with similar syntactic but fine-grained semantic differences due to word-level distinctions. For each prompt pair, we instruct text-to-image models to generate the image pairs and evaluate the text-image consistency scores $(s^1, s^2)$. We then calculate the arithmetic mean score as $s_a = (s^1 + s^2)/2$, and the geometric mean score as $s_g = \sqrt{s^1 * s^2}$

Ideally, models should precisely distinguish the semantic nuances between prompts and accurately generate the corresponding images. However, even for the SOTA AR-related model (Chen et al., 2025b; Wang et al., 2024), the geometric mean in PairComp is significantly lower than the arithmetic mean (Figure 1.a). Considering that the geometric mean is highly sensitive to lower values, the results indicate the instability control over fine-grained visual generation. The examples in Figure 1.b further illustrate the inability to accurately control details such as color and spatial relationship. We argue that this problem lies in the lack of fine-grained text-image semantic alignment. Image–text alignment training does not provide fine-grained annotations that specify which part of a sentence corresponds to which region in an image. Moreover, images often contain irrelevant low-level semantics not mentioned in the text (Ge et al., 2023), which further introduces erroneous biases in fine-grained semantics, leading some text tokens to form incorrect alignments with certain visual details.

Thus, a crucial question emerges: ***How can we achieve robust fine-grained text-image alignment to enable precise control over visual semantics in AR-based text-to-image generation?*** Some stud-

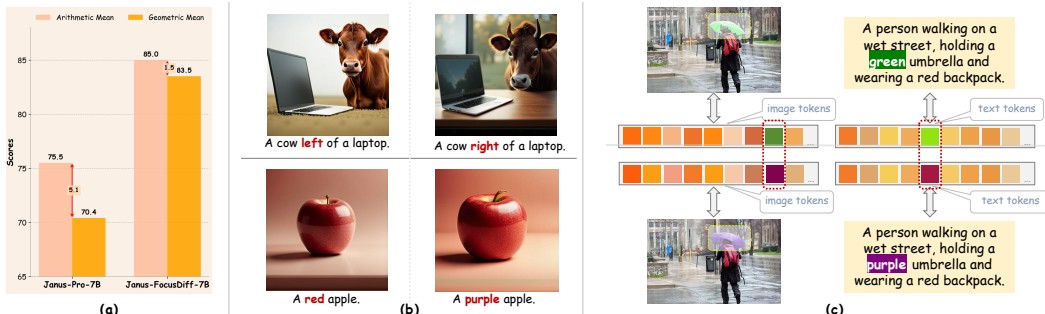

Figure 1: (a) geometric/arithmetic mean score in PairComp for AR-related models. (b) Examples of Janus-Pro failing to generate images precisely aligned with the prompt. (c) The subtle sensory differences between images or between texts result in only minor alterations to specific tokens.

ies (Yin et al., 2024; Zhao et al., 2024b) in multimodal comprehension leverage contrastive learning to build extra constraints for intra-sequence fine-grained token embedding alignment. However, they undermine the core design philosophy of the decoder-only AR – the causal dependency of tokens, failing to fully leverage the successful infrastructure of LLMs. We aim to find an elegant solution for fine-grained text-image alignment without altering the original AR-based training paradigm.

We introduce **FocusDiff**, a method that enhances fine-grained text-image semantic alignment by learning from the differences between similar text-image pairs, without disrupting the original AR-based training paradigm. Specifically, **from the data perspective**, we introduce `FocusDiff-Data`, expanding the training case from a single text-image pair $\{(\mathcal{T}, \mathcal{I})\}$ into a set of two pairs $\{(\mathcal{T}^1, \mathcal{I}^1, \mathcal{T}^2, \mathcal{I}^2)\}$. Here, $\mathcal{T}^1$ and $\mathcal{T}^2$, as well as $\mathcal{I}^1$ and $\mathcal{I}^2$, appear similar in overall expression but differ in fine-grained details, with $\mathcal{T}^1$ being consistent with $\mathcal{I}^1$ but not with $\mathcal{I}^2$, and vice versa. As shown in Figure 1.c, the subtle sensory differences between images or between texts result in only minor alterations to specific visual or textual tokens. Therefore, by comparing the token differences between these pairs, Multimodal Large Language Model (MLLM) can trace how changes in text tokens lead to specific changes in visual tokens, establishing fine-grained semantic associations between the two modalities.

**From the training perspective**, we introduce Pair-GRPO, a reinforcement learning (RL) method that guides the model in learning fine-grained semantic differences through an exploration-exploitation trade-off. We formulate image generation as a Markov decision process and extend the Group Relative Policy Optimization (GRPO) framework (Shao et al., 2024) to visual generation with a QA-based reward model, which eliminates the value function and estimates advantages in a group-relative manner. We make two key improvements:

**(1) Expanding the Group Concept:** While vanilla GRPO considers $G$ responses from the same prompt as a group, we expand this to include $2 \times G$ responses from pairs of similar prompts with fine-grained semantic differences from `FocusDiff-Data`.

**(2) Shifting Focus from Exploitation to Exploration:** Unlike vanilla GRPO, which encourages fully autonomous exploration without ground-truth images, we provide ground-truth images from `FocusDiff-Data` during early training to enhance exploration and guide the model to better grasp fine-grained semantic differences. As training progresses, we gradually reduce the reliance on these ground-truth images, transitioning from exploitation-first to exploration-first.

Thanks to our novel training data and training strategy, with Janus-Pro as the backbone, we realize better fine-grained text-image semantic alignments and achieve precise control over visual semantics during text-to-image generation. Our main contributions are threefold:

- We introduce PairComp benchmark, featuring test cases with two prompts only differing in fine-grained semantics, highlighting existing models' limitations in precise visual control.

- We propose FocusDiff, a paired text-image dataset with an improved GRPO-based training paradigm, focusing on fine-grained semantic differences to boost text-image alignment.

- We achieve superior performance on existing text-to-image benchmarks and significantly outperform most prior prominent methods on PairComp.

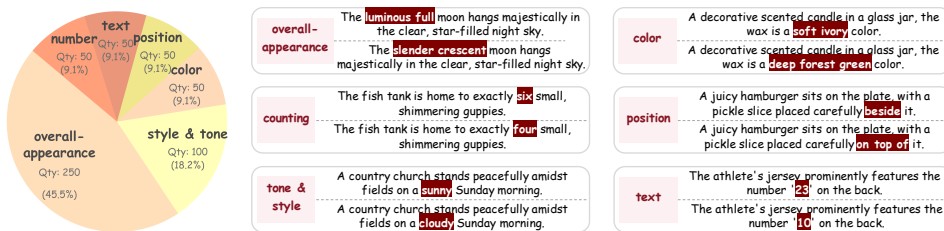

Figure 2: Statistical information of PairComp and test case examples for each subtask.

# 2 BENCHMARK: PAIRCOMP

**Data format and Task Categorization.** In traditional text-to-image benchmarks (Ghosh et al., 2023; Hu et al., 2024), each test case consists of a single prompt, which is used to measure the overall semantic alignment between the prompt and the generated image. In this section, we introduce a new benchmark called **PairComp**. Each test case in PairComp contains two similar prompts with subtle differences. By comparing the accuracy of the images generated by the model for each prompt, we evaluate whether the model has focused on the fine-grained semantic differences in the prompts to produce the corresponding correct images. The two prompts in a test case exhibit word-level differences leading to noticeable distinctions in certain fine-grained semantic aspects, which can be categorized into six types: (1) Overall appearance; (2) Color; (3) Counting; (4) Position; (5) Style & Tone; (6) Text. In Figure 2, we present examples of each category. See more details in Appendix B.

**Evaluation Protocols.** We use InternVL2.5-26B (Chen et al., 2024) as the primary evaluation model to assess the semantic alignment between the generated images and the prompts. Specifically, for each image-prompt pair, we query the model with "Does this image match the description? Please directly respond with yes or no." We record the probability of the model responding with "yes" ("no") as $P_{yes}$ ($P_{no}$). And the semantic alignment score is calculated as $S(\mathcal{I}, \mathcal{T}) = P_{yes}/(P_{yes} + P_{no})$.

On this basis, given a subtask $\{(\mathcal{T}^{1,i}, \mathcal{T}^{2,i})\}$, for each prompt pair, we instruct a text-to-image model to generate corresponding images $\{\{(\mathcal{T}^{1,i}, \mathcal{I}_k^{1,i})\}_{k=1}^K, \{(\mathcal{T}^{2,i}, \mathcal{I}_k^{2,i})\}_{k=1}^K\}_{i=1}^N$, with each prompt generating $K = 2$ images. We define $s_k^{j,i} = S(\mathcal{I}_k^{j,i}, \mathcal{T}^{j,i})$, introduce two evaluation metrics: arithmetic mean $s_a = \frac{1}{4N} \sum_{i=1}^N \sum_{j=1}^2 \sum_{k=1}^2 s_k^{j,i}$, and geometric mean $s_g = \frac{1}{N} \sum_{i=1}^N \sqrt[4]{\prod_{j=1}^2 \prod_{k=1}^2 s_k^{j,i}}$. Here, $s_a$ measures the overall semantic alignment of the generated images with the prompts, while $s_g$ assesses the model's fine-grained precision and stability in generating images for similar prompts. $\mathcal{T}^{j,i}$ means the $j$-th prompt in $i$-th test case, where $j = 0$ or $1$ and $i \in [1, N]$, with $N$ being the total number of test cases in the task. Similarly, $\mathcal{I}_k^{j,i}$ means the $k$-th image generated for the $j$-th prompt in $i$-th test case, where $k \in [1, K]$ with $K$ being the total number of images generated for one prompt.

# 3 METHOD: FOCUSDIFF

In this section, we introduce FocusDiff, a novel text-to-image method that focuses on the differences between similar text-image pairs to enhance fine-grained text-image alignment. From the data perspective, we propose `FocusDiff-Data`, expanding the training dataset from a single text-image pair to a set of two pairs. From the training perspective, we further propose Pair-GRPO, an improved RL framework that guides the model to better focus on fine-grained semantic differences.

## 3.1 DATA PERSPECTIVE: FOCUSDIFF-DATA

Traditional text-to-image training data comprises isolated text-image pairs lacking explicit connections. While ensuring global alignment, it often fails to achieve fine-grained alignment without detailed annotations indicating which portion of a sentence aligns with which region of an image. As images often contain low-level information not mentioned in the text; even when the text does offer fine-grained descriptions of the image, this "redundant" information can act as a confounder, misleading the model about which visual region should truly correspond to each fine-grained description.

To address this issue, we turn to differential learning, which expands a single text-image pair $\{(\mathcal{T}, \mathcal{I})\}$ into two pairs $\{(\mathcal{T}^1, \mathcal{I}^1, \mathcal{T}^2, \mathcal{I}^2)\}$. While $\mathcal{T}^1$ and $\mathcal{T}^2$, as well as $\mathcal{I}^1$ and $\mathcal{I}^2$, are similar in overall

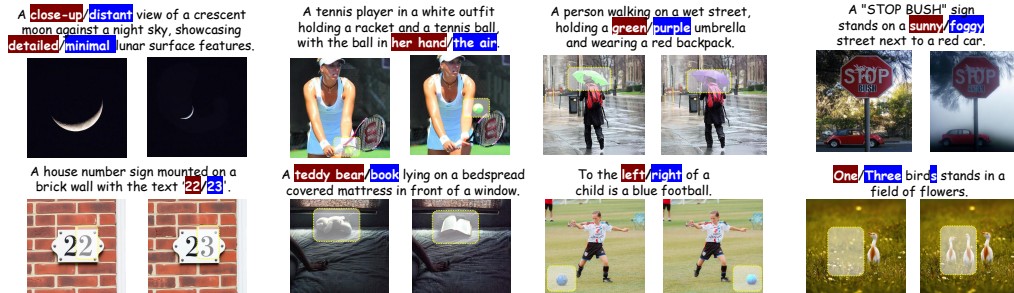

Figure 3: Examples of training data in `FocusDiff-Data`.

expression and global semantics, they differ in fine-grained details. Consequently, $\mathcal{T}^1$ is semantically aligned with $\mathcal{I}^1$ but not with $\mathcal{I}^2$, and vice versa. In an AR framework such as Janus-Pro, with text and images represented as tokens, only a few token-level differences exist between $\mathcal{T}^1$ and $\mathcal{T}^2$, as well as between $\mathcal{I}^1$ and $\mathcal{I}^2$. Then, the model is able to deduce how changes in text tokens lead to specific changes in visual tokens, allowing it to focus on subtle differences between texts and images, which ultimately enhances fine-grained text-image semantic alignment.

To obtain such paired data, especially pairs of similar images, we turn to image editing datasets (Yu et al., 2024; Zhao et al., 2024a), which involve before-and-after-editing image pairs in which only localized regions are modified. We collect numerous image pairs, covering a diverse range of editing types to reflect differences in various attributes. And then we can employ a powerful visual comprehension model to generate style-similar captions for each pair.

Specifically, given the subpar quality of existing image editing datasets, we perform an initial screening to assess three key aspects for each case: (1) editing instructions following, (2) non-edited areas preserving, and (3) natural appearance. After eliminating substandard samples, we input the before-and-after image pair and the editing instruction into InternVL2.5-26B, prompting it to generate captions with similar structure but different key words to highlight the subtle image differences.

After generating the captions $(\mathcal{T}^1, \mathcal{T}^2)$ for the images $(\mathcal{I}^1, \mathcal{I}^2)$, we then perform a post-verification to ensure three conditions: (1) $\mathcal{T}^1$ and $\mathcal{T}^2$ exhibit similar semantic structures; (2) $\mathcal{T}^1$ is semantically aligned with $\mathcal{I}^1$, and $\mathcal{T}^2$ with $\mathcal{I}^2$; (3) $\mathcal{T}^1$ is not semantically aligned with $\mathcal{I}^2$, nor $\mathcal{T}^2$ with $\mathcal{I}^1$. If any condition is violated, we leverage InternVL2.5-26B to regenerate captions and re-verify.

Ultimately, we retained around $200,000$ high-quality data pairs. Randomly selected examples from `FocusDiff-Data` are visualised in Fig. 3, where the images and their corresponding prompts exhibit only region-level or word-level differences. See more details in Appendix C.

## 3.2 Training Perspective: Pair-GRPO

With `FocusDiff-Data`, we first conduct a supervised text-to-image fine-tuning. Subsequently, we perform reinforcement learning based on an improved version of GRPO (Shao et al., 2024) (Figure 4) on text-to-image generation, realizing a better balance of exploration-exploitation trade-off.

**Vanilla GRPO with QA-based Reward.** We aim to adopt GRPO as the framework for reinforcement learning, which enhances Proximal Policy Optimization (PPO) by eliminating the value function and estimating the advantages in a group-relative manner. Specifically, for a specific prompt, the old policy first samples a group of $G$ individual images as the response group $\mathcal{G} = \{\mathcal{I}_k^1\}_{k=1}^G$. We input each response within the group into the reward function to obtain the individual reward $\mathrm{R}_{\mathcal{I}_k}$, with the advantages $A_k = \frac{\mathrm{R}_{\mathcal{I}_k} - \mathrm{mean}\left(\{\mathrm{R}_{\mathcal{I}_k}\}_{k=1}^G\right)}{\mathrm{std}\left(\{\mathrm{R}_{\mathcal{I}_k}\}_{k=1}^G\right)}$ measuring the relative quality of output compared to the average reward. Then, we update the policy network parameters by the following training loss:

$$\mathcal{J}(\theta) = \mathbb{E}_{\substack{(\mathcal{T},a)\sim\mathcal{D} \\ \{y_k\}_{k=1}^G \sim \pi_{\theta_{\mathrm{old}}}(\cdot|\mathcal{T})}} \left[ \frac{1}{\sum_{k=1}^G |y_k|} \sum_{k=1}^G \sum_{j=1}^{|y_k|} \left( \min\left(\rho_{k,j}A_k, \mathrm{clip}\left(\rho_{k,j}, 1-\varepsilon, 1+\varepsilon\right)A_k\right) - \beta D_{\mathrm{KL}} \right) \right],$$
(1)

$\rho_{k,j} = \frac{\pi_\theta(y_{k,j}|\mathcal{T}, y_{k,<j})}{\pi_{\theta_{\mathrm{old}}}(y_{k,j}|\mathcal{T}, y_{k,<j})}$ is the ratio between probabilities of $\pi_\theta$ and $\pi_{\theta_{\mathrm{old}}}$ for outputting current token.

Table 1: Main results on PairComp with InternVL2.5-26B as the evaluation model. The best results are in **bold fonts** with the second best underlined.

| Method | #Params | Overall Appear. $s_a \uparrow$ | $s_g \uparrow$ | Color $s_a \uparrow$ | $s_g \uparrow$ | Counting $s_a \uparrow$ | $s_g \uparrow$ | Position $s_a \uparrow$ | $s_g \uparrow$ | Style&Tone $s_a \uparrow$ | $s_g \uparrow$ | Text $s_a \uparrow$ | $s_g \uparrow$ | **Average** $s_a \uparrow$ | $s_g \uparrow$ |
|---|---|---|---|---|---|---|---|---|---|---|---|---|---|---|---|
| *Diffusion-Only* | | | | | | | | | | | | | | | |
| SD3 (Esser et al., 2024) | 8B | 82.5 | 77.0 | 95.4 | 94.6 | 74.0 | 70.3 | 71.9 | 68.5 | 89.4 | 86.2 | 93.1 | 92.0 | 84.4 | 81.4 |
| FLUX.1-dev (Labs, 2024) | 12B | 78.7 | 71.1 | 94.3 | 92.0 | 63.9 | 60.0 | 70.4 | 66.1 | 84.4 | 79.7 | 90.1 | 85.4 | 80.3 | 75.7 |
| Sana-1.5 (Xie et al., 2025) | 4.8B | 83.8 | 79.5 | 97.3 | 96.8 | 74.1 | 71.5 | 69.1 | 64.0 | 92.7 | 90.1 | 82.4 | 77.9 | 83.2 | 80.0 |
| Janus-Flow (Ma et al., 2024) | 1.3B | 62.1 | 54.0 | 74.1 | 67.5 | 45.0 | 40.2 | 45.7 | 36.8 | 84.4 | 80.3 | 21.7 | 15.2 | 55.5 | 49.0 |
| Flow-GRPO (Liu et al., 2025) | 2.5B | 82.3 | 77.8 | 95.3 | 95.2 | 61.5 | 57.2 | 70.6 | 67.0 | 90.1 | 88.1 | 93.2 | 92.1 | 82.2 | 79.6 |
| Lumina-Image2.0 (Qin et al., 2025) | 2B | 77.5 | 72.6 | 94.1 | 93.8 | 63.7 | 60.3 | 69.8 | 67.5 | 91.1 | 88.9 | 59.1 | 49.9 | 75.9 | 72.2 |
| HiDream-I1-Full (Cai et al., 2025) | 17B | 82.9 | 77.8 | 95.8 | 95.0 | 71.2 | 69.1 | 71.0 | 67.0 | 91.3 | 89.2 | 96.9 | 95.3 | 84.9 | 82.2 |
| HunyuanImage-2.1 (Team, 2025) | 17B | 85.3 | 81.4 | 97.4 | 96.0 | 82.5 | 81.1 | 75.3 | 73.7 | 92.5 | 91.5 | 62.0 | 54.0 | 82.5 | 79.6 |
| Qwen-Image (Wu et al., 2025a) | 27B | 85.2 | 80.8 | 97.8 | 97.4 | 83.9 | 83.3 | 78.9 | 76.7 | 94.2 | 93.2 | 97.7 | 97.4 | 89.6 | 88.1 |
| *Hybrid Model (AR + Diffusion)* | | | | | | | | | | | | | | | |
| Show-o (Xie et al., 2024) | 1.3B | 68.5 | 62.2 | 87.2 | 85.0 | 58.2 | 55.2 | 45.2 | 40.6 | 87.8 | 84.7 | 34.9 | 26.8 | 63.6 | 59.1 |
| SEED-X (Ge et al., 2024) | 17B | 83.2 | 79.7 | 95.5 | 94.5 | 64.9 | 62.3 | 63.0 | 59.9 | 90.0 | 87.3 | 52.2 | 45.1 | 74.8 | 71.5 |
| BLIP3-o (Chen et al., 2025a) | 8B | 83.4 | 78.7 | 95.8 | 94.2 | 68.2 | 65.7 | 72.5 | 69.2 | 93.4 | 91.4 | 62.2 | 53.6 | 79.3 | 75.5 |
| Omni-Gen2 (Wu et al., 2025b) | 7B | 80.6 | 75.2 | 94.6 | 93.6 | 65.9 | 61.8 | 71.6 | 68.1 | 89.7 | 87.0 | 92.4 | 89.9 | 82.5 | 79.4 |
| Bagel-Think (Deng et al., 2025) | 14B | 81.8 | 78.1 | 90.2 | 88.7 | 69.2 | 66.6 | 68.8 | 62.6 | 87.9 | 84.2 | 60.2 | 53.1 | 76.4 | 72.2 |
| X-Omni-En (Geng et al., 2025) | 9.6B | 78.1 | 72.4 | 92.3 | 91.0 | 68.5 | 66.5 | 62.7 | 58.1 | 85.8 | 82.4 | 90.5 | 88.9 | 79.7 | 76.6 |
| *AR-Only* | | | | | | | | | | | | | | | |
| LLamaGen (Sun et al., 2024) | 3.1B | 53.5 | 45.4 | 67.0 | 61.2 | 45.3 | 39.5 | 42.1 | 35.4 | 68.8 | 60.1 | 18.0 | 12.0 | 49.1 | 42.3 |
| VILA-U (Wu et al., 2024) | 7B | 70.5 | 65.0 | 82.9 | 79.5 | 53.3 | 48.6 | 53.4 | 46.0 | 86.6 | 83.2 | 30.9 | 25.7 | 62.9 | 58.0 |
| Emu3 (Wang et al., 2024) | 8B | 73.8 | 66.1 | 87.3 | 85.0 | 60.4 | 57.2 | 55.1 | 49.8 | 85.9 | 82.2 | 48.5 | 39.1 | 68.5 | 63.2 |
| Infinity (Han et al., 2024) | 8B | 79.5 | 73.2 | 93.7 | 92.1 | 65.5 | 62.1 | 62.7 | 57.5 | 87.1 | 83.0 | 73.5 | 68.2 | 77.0 | 72.7 |
| Janus-Pro-1B (Chen et al., 2025b) | 1B | 75.6 | 69.5 | 89.7 | 87.7 | 36.1 | 29.5 | 56.2 | 50.2 | 92.3 | 90.4 | 37.6 | 28.0 | 64.6 | 59.2 |
| Janus-Pro-7B (Chen et al., 2025b) | 7B | 82.3 | 75.6 | 95.7 | 94.0 | 52.7 | 47.1 | 69.4 | 63.9 | 92.0 | 88.7 | 60.8 | 53.2 | 75.5 | 70.4 |
| Lumina-mGPT 2.0 (Xin et al., 2025) | 7B | 71.8 | 64.2 | 86.0 | 82.5 | 54.8 | 51.6 | 53.8 | 47.1 | 86.8 | 83.0 | 38.0 | 27.3 | 65.2 | 59.3 |
| T2I-R1 (Jiang et al., 2025) | 7B | 84.6 | 80.3 | 96.5 | 95.9 | 68.1 | 65.2 | 71.3 | 67.5 | 91.2 | 89.2 | 82.5 | 77.5 | 82.4 | 79.3 |
| Janus-Pro-R1 (Pan et al., 2025) | 7B | 84.1 | 79.9 | 96.7 | 95.9 | 68.6 | 65.8 | 71.9 | 70.0 | 93.3 | 91.8 | 77.1 | 71.6 | 82.0 | 79.2 |
| **Janus-FocusDiff-1B** | 1B | 78.4 | 75.0 | 91.7 | 90.0 | 49.5 | 44.4 | 64.7 | 61.8 | 91.9 | 90.8 | 49.8 | 46.3 | 71.0 | 68.1 |
| Janus-Pro-7B + SFT | 7B | 82.6 | 76.6 | 96.1 | 94.8 | 58.4 | 53.5 | 71.0 | 65.4 | 92.6 | 89.6 | 64.8 | 58.1 | 77.6 | 73.0 |
| **Janus-FocusDiff-7B** | 7B | 85.4 | 82.4 | 97.8 | 97.7 | 71.0 | 69.0 | 75.9 | 74.0 | 94.3 | 93.9 | 85.3 | 83.8 | 85.0 | 83.5 |

As for the reward model, the overall design philosophy is to leverage a QA-based visual comprehension model (*i.e.*, InternVL2.5-26B) to provide appropriate incentives, which will return a consistency score $R_{\mathcal{I}} \in [0, 1]$ for each text-image pair. More details are provided in Appendix A.

**Pair-GRPO for Fine-Grained Semantic Focusing.** To enhance the model's ability to capture subtle differences between two prompts, we extend the group concept in GRPO from images generated by a single prompt to those generated by pairs of similar prompts. This aligns with our core idea of comparing the outputs of similar prompt pairs. Specifically, give a pair of input prompt $\{\mathcal{T}^1, \mathcal{T}^2\}$ with similar global expressions but fine-grained semantics differences, a group of $2G$ images $\{\mathcal{I}_k^1\}_{k=1}^G$ for $\mathcal{T}^1$ and $\{\mathcal{I}_k^2\}_{k=1}^G$ for $\mathcal{T}^2$ are sampled from the old policy. And then $\{\mathcal{I}_k^1\}_{k=1}^G$ and $\{\mathcal{I}_k^2\}_{k=1}^G$ are assigned to the same group $\mathcal{G}_0 = \{(\mathcal{T}^1, \mathcal{I}_k^1)\}_{k=1}^G \cup \{(\mathcal{T}^2, \mathcal{I}_k^2)\}_{k=1}^G$ for advantage calculation.

Furthermore, from the `FocusDiff-Data` dataset, we could also obtain the ground-truth images $\hat{\mathcal{I}}^1$ and $\hat{\mathcal{I}}^2$ corresponding to $\mathcal{T}^1$ and $\mathcal{T}^2$. Despite the high similarity between $\hat{\mathcal{I}}^1$ and $\hat{\mathcal{I}}^2$, during construction we ensure that $\hat{\mathcal{I}}^1$ achieves a favorable reward score when conditioned on $\mathcal{T}^1$, but achieves an unfavorable score when conditioned on $\mathcal{T}^2$. Thus, if we further incorporate $\hat{\mathcal{I}}^1$ into the group, it can assume a dual role within the group: it serves as a positive guide in $\{(\mathcal{T}^1, \mathcal{I}_k^1)\}_{k=1}^G$ indicating to the model about the correct visual semantics, and as a cautionary counterexample in $\{(\mathcal{T}^2, \mathcal{I}_k^2)\}_{k=1}^G$, warning the model to avoid generating erroneous visual semantics that are commonly encountered. The same applies to $\hat{\mathcal{I}}^2$.

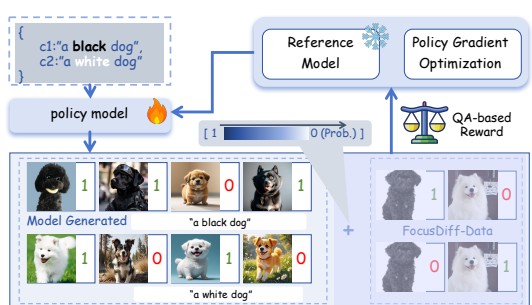

Figure 4: The framework of Pair-GRPO.

On this basis, we introduce a dynamic probability $p$ that starts at $1.0$ and gradually decreases to $0.0$ during RL training. At each training iteration, with probability $p$, we expand the group $\mathcal{G}$ to include the above additional pairs from `FocusDiff-Data`: $\mathcal{G} = \mathcal{G}_0 + \{(\mathcal{T}^1, \hat{\mathcal{I}}^1), (\mathcal{T}^1, \hat{\mathcal{I}}^2), (\mathcal{T}^2, \hat{\mathcal{I}}^1), (\mathcal{T}^2, \hat{\mathcal{I}}^2)\}$. Otherwise, the group remains as $\mathcal{G} = \mathcal{G}_0$. **This is a process of shifting focus from exploitation to exploration.** In the early stages of training, the labeled images from the dataset encourage more exploitation to the model, offering more appropriate guidance. As training progresses and the model's ability to grasp fine-grained differences strengthens,

Table 2: Main results on PairComp with Qwen2.5-VL-72B as the evaluation model. The best results are in **bold fonts** with the second best underlined.

| Method | #Params | Overall Appear. | | Color | | Counting | | Position | | Style&Tone | | Text | | **Average** | |
|---|---|---|---|---|---|---|---|---|---|---|---|---|---|---|---|
| | | $s_a \uparrow$ | $s_g \uparrow$ | $s_a \uparrow$ | $s_g \uparrow$ | $s_a \uparrow$ | $s_g \uparrow$ | $s_a \uparrow$ | $s_g \uparrow$ | $s_a \uparrow$ | $s_g \uparrow$ | $s_a \uparrow$ | $s_g \uparrow$ | $s_a \uparrow$ | $s_g \uparrow$ |
| *Diffusion-Only* | | | | | | | | | | | | | | | |
| SD3 (Esser et al., 2024) | 8B | 73.0 | 64.3 | 94.1 | 91.7 | 54.0 | 44.8 | 50.0 | 38.3 | 86.4 | 79.4 | 87.6 | 81.8 | 74.2 | 66.7 |
| FLUX.1-dev (Labs, 2024) | 12B | 68.9 | 59.1 | 92.3 | 87.7 | 44.2 | 34.4 | 50.0 | 40.9 | 82.5 | 72.6 | 87.3 | 81.5 | 70.9 | 62.7 |
| FLUX.1-dev-PrefGRPO (Wang et al., 2025) | 12B | 73.0 | 65.1 | 95.1 | 92.2 | 46.0 | 38.8 | 52.4 | 45.6 | 84.2 | 77.8 | 87.6 | 83.7 | 73.1 | 67.2 |
| Sana-1.5 (Xie et al., 2025) | 4.8B | 72.8 | 64.2 | 95.1 | 92.8 | 45.6 | 37.7 | 47.7 | 36.0 | 91.8 | 87.5 | 67.1 | 59.5 | 70.0 | 63.0 |
| Janus-Flow (Ma et al., 2024) | 1.3B | 47.8 | 30.2 | 66.8 | 50.1 | 22.7 | 12.1 | 23.7 | 7.4 | 81.7 | 70.8 | 16.2 | 8.0 | 43.2 | 29.8 |
| Flow-GRPO (Liu et al., 2025) | 2.5B | 73.2 | 64.1 | 94.3 | 92.0 | 47.1 | 33.5 | 49.4 | 37.3 | 87.9 | 82.6 | 88.0 | 82.3 | 73.3 | 65.3 |
| Lumina-Image 2.0 (Qin et al., 2025) | 2B | 66.5 | 55.5 | 87.3 | 82.1 | 39.8 | 23.4 | 49.0 | 36.8 | 89.1 | 83.7 | 46.5 | 31.8 | 63.0 | 52.2 |
| HiDream-I1-Full (Cai et al., 2025) | 17B | 72.6 | 64.3 | 92.4 | 90.5 | 46.9 | 40.4 | 51.6 | 42.1 | 88.9 | 83.6 | 93.8 | 89.8 | 74.4 | 68.5 |
| HunyuanImage-2.1 (Team, 2025) | 17B | 72.2 | 64.3 | 84.1 | 80.2 | 67.0 | 61.0 | 64.6 | 54.0 | 85.3 | 82.1 | 45.8 | 36.3 | 69.8 | 63.0 |
| Qwen-Image (Wu et al., 2025a) | 27B | 73.6 | 70.1 | 94.7 | 93.8 | 68.3 | 64.3 | 66.2 | 51.8 | 93.9 | 91.0 | 94.7 | 93.7 | 81.9 | 77.5 |
| *Hybrid Model (AR+Diffusion)* | | | | | | | | | | | | | | | |
| Show-o (Xie et al., 2024) | 1.3B | 50.3 | 35.3 | 75.0 | 65.7 | 39.1 | 28.3 | 23.8 | 11.9 | 82.9 | 74.8 | 24.0 | 14.8 | 49.2 | 38.4 |
| SEED-X (Ge et al., 2024) | 17B | 71.4 | 58.9 | 90.4 | 88.0 | 29.6 | 22.6 | 40.6 | 26.0 | 87.9 | 82.1 | 32.4 | 22.8 | 58.7 | 50.1 |
| BLIP3-o (Chen et al., 2025a) | 8B | 70.8 | 61.6 | 90.7 | 86.5 | 41.6 | 35.2 | 50.9 | 38.0 | 91.0 | 86.9 | 48.7 | 34.3 | 65.6 | 57.1 |
| Omni-Gen2 (Wu et al., 2025b) | 7B | 72.7 | 62.5 | 91.1 | 86.0 | 40.9 | 32.5 | 52.1 | 44.6 | 87.9 | 81.6 | 79.2 | 71.5 | 70.7 | 63.1 |
| Bagel-Think (Deng et al., 2025) | 14B | 70.5 | 59.7 | 89.1 | 85.3 | 45.9 | 36.2 | 48.1 | 40.1 | 84.9 | 75.2 | 43.9 | 32.1 | 63.7 | 54.8 |
| X-Omni-En (Geng et al., 2025) | 9.6B | 67.4 | 59.2 | 86.5 | 78.8 | 36.5 | 28.2 | 45.7 | 37.6 | 82.4 | 74.9 | 79.7 | 71.5 | 66.4 | 58.4 |
| *AR-only* | | | | | | | | | | | | | | | |
| LLamaGen (Sun et al., 2024) | 3.1B | 33.1 | 16.6 | 45.6 | 31.5 | 19.2 | 9.6 | 19.4 | 9.7 | 56.6 | 38.8 | 6.2 | 1.8 | 30.0 | 18.0 |
| VILA-U (Wu et al., 2024) | 7B | 56.3 | 41.2 | 75.9 | 67.1 | 25.6 | 14.5 | 28.3 | 15.8 | 86.3 | 79.3 | 17.5 | 10.6 | 48.3 | 38.1 |
| Emu3 (Wang et al., 2024) | 8B | 62.5 | 49.3 | 79.0 | 68.2 | 40.8 | 29.7 | 37.9 | 29.2 | 84.1 | 75.3 | 30.3 | 19.1 | 55.8 | 45.1 |
| Infinity (Han et al., 2024) | 8B | 68.0 | 52.3 | 91.0 | 86.9 | 29.2 | 18.6 | 37.7 | 19.7 | 85.0 | 76.6 | 55.8 | 43.9 | 61.1 | 49.7 |
| Janus-Pro-1B (Chen et al., 2025b) | 1B | 62.1 | 46.6 | 85.9 | 81.7 | 20.5 | 13.1 | 36.3 | 19.2 | 90.0 | 85.0 | 31.3 | 22.5 | 54.3 | 44.7 |
| Janus-Pro-7B (Chen et al., 2025b) | 7B | 70.0 | 60.6 | 91.0 | 88.6 | 29.3 | 19.4 | 48.0 | 29.1 | 89.5 | 85.0 | 53.8 | 41.7 | 63.6 | 54.1 |
| Lumina-mGPT 2.0 (Xin et al., 2025) | 7B | 62.9 | 52.0 | 82.4 | 73.8 | 32.6 | 24.8 | 35.4 | 26.9 | 82.0 | 72.3 | 32.8 | 20.9 | 54.7 | 45.1 |
| T2I-R1 (Jiang et al., 2025) | 7B | 73.1 | 65.8 | 94.7 | 91.9 | 41.9 | 32.6 | 51.2 | 39.6 | 91.3 | 86.8 | 70.4 | 62.5 | 70.4 | 63.2 |
| Janus-Pro-R1 (Pan et al., 2025) | 7B | 72.2 | 64.2 | 95.0 | 91.9 | 42.7 | 34.6 | 52.0 | 43.1 | 93.2 | 87.7 | 65.5 | 56.9 | 70.1 | 63.1 |
| **Janus-FocusDiff-1B** | 1B | 67.0 | 57.9 | 90.0 | 86.0 | 25.6 | 20.1 | 42.3 | 32.5 | 91.6 | 86.9 | 38.7 | 31.4 | 59.2 | 52.5 |
| Janus-Pro-7B + SFT | 7B | 70.7 | 61.8 | 92.2 | 91.0 | 36.9 | 28.1 | 52.5 | 35.0 | 90.0 | 85.8 | 58.2 | 46.7 | 66.8 | 58.1 |
| **Janus-FocusDiff-7B** | 7B | 73.8 | 67.5 | 96.2 | 95.0 | 46.2 | 39.7 | 60.6 | 53.4 | 94.1 | 91.1 | 77.8 | 71.4 | 74.8 | 69.9 |

the probability of providing labeled images gradually decreases. And finally, we encourage model to develop advanced problem-solving strategies through fully autonomous exploration.

In each iteration, after defining the group concept, we employ the same way as vanilla GRPO to calculate the rewards, advantages and the objective function.

## 4 EXPERIMENTS

We employ Janus-Pro (Chen et al., 2025b) as the backbone, developing Janus-FocusDiff, excelling in text-to-image generation, with improved capabilities of vision-language alignment. More details are given in Appendix D and E.

### 4.1 MAIN RESULTS ON PAIRCOMP

We conduct zero-shot evaluations on PairComp for our model and recent advanced diffusion-based, AR-based and hybrid text-to-image methods. Following the evaluation protocols in § 2, we report the arithmetic mean scores $s_a$ and geometric mean scores $s_g$ in Table 1 and Table 2. We leverage InternVL2.5-26B and Qwen2.5-VL-72B as the evaluators. From Table 1, we have the following key findings of existing methods:

(1) **The overall text-image alignment is satisfactory.** The leading models of each baseline type – Qwen-Image among diffusion-based models, Omni-Gen2 among hybrid models, and T2I-R1 among AR-based models – all exhibit relatively strong arithmetic-mean scores. Especially Qwen-Image, a 27B model including a 20B DiT, ranks far ahead on the benchmark.

(2) **The stability of image generation is poor,** making it difficult to precisely control fine-grained visual semantics that reflect subtle differences specified in the prompts. The gap between the geometric mean and the arithmetic mean reflects the stability of a model's image generation. It can be seen that most methods (except Qwen-Image) struggle to achieve ideal geometric mean scores, indicating poor stability in image generation. For example, the average $s_g$ of FLUX is 4.6 points lower than its $s_a$, and the average $s_g$ of Janus-Pro-7B is 5.1 points lower than its $s_a$. Besides, AR-based methods exhibit slightly lower stability in image quality compared to diffusion-based methods.

Compared to existing methods, Janus-FocusDiff-7B achieves the following advantages: (1) **Improved text-image alignment** is achieved with higher arithmetic mean scores. After training,

Table 3: Comparison with various leading models on GenEval, T2I-CompBench and DPG-Bench on zero-shot text-to-image generation. The best results are in **bold fonts** with the second best underlined.

| Method | #Params | GenEval | | | | | | | T2I-CompBench | | | DPG-Bench |
|---|---|---|---|---|---|---|---|---|---|---|---|---|
| | | Overall↑ | SingObj↑ | TwoObj↑ | Counting↑ | Color↑ | Pos.↑ | ColorAttr↑ | Color↑ | Shape↑ | Texture↑ | Avg↑ |
| *Diffusion-Only* | | | | | | | | | | | | |
| SD3 (Esser et al., 2024) | 8B | 0.74 | 0.99 | 0.94 | 0.72 | 0.89 | 0.33 | 0.60 | - | - | - | 84.08 |
| FLUX.1-dev (Labs, 2024) | 12B | 0.66 | 0.98 | 0.79 | 0.73 | 0.77 | 0.22 | 0.45 | - | - | - | 83.79 |
| Sana-1.5 (Xie et al., 2025) | 4.8B | 0.81 | 0.99 | 0.93 | 0.86 | 0.84 | 0.59 | 0.65 | - | - | - | 84.70 |
| Janus-Flow (Ma et al., 2024) | 1.3B | 0.63 | 0.97 | 0.59 | 0.45 | 0.83 | 0.53 | 0.42 | - | - | - | 80.09 |
| Lumina-Image 2.0 (Qin et al., 2025) | 2B | 0.78 | 0.99 | 0.87 | 0.67 | 0.86 | 0.70 | 0.62 | 82.1 | 60.3 | 74.2 | 82.32 |
| HunyuanImage-2.1 (Team, 2025) | 17B | 0.79 | 0.98 | 0.92 | 0.71 | 0.86 | 0.66 | 0.61 | 76.9 | 59.3 | 74.3 | 85.15 |
| Qwen-Image (Wu et al., 2025a) | 27B | **0.91** | **1.00** | 0.95 | **0.93** | 0.92 | **0.87** | **0.83** | **85.1** | **63.1** | **76.5** | **88.32** |
| *Hybrid Model (AR + Diffusion)* | | | | | | | | | | | | |
| Show-o (Xie et al., 2024) | 1.3B | 0.68 | 0.98 | 0.80 | 0.66 | 0.84 | 0.31 | 0.50 | 56.0 | 41.0 | 46.0 | 67.48 |
| Show-o+PARM (Guo et al., 2025b) | 1.3B | 0.69 | 0.97 | 0.75 | 0.60 | 0.83 | 0.54 | 0.53 | 75.0 | 56.0 | 66.0 | - |
| SEED-X (Ge et al., 2024) | 17B | 0.49 | 0.96 | 0.57 | 0.29 | 0.82 | 0.14 | 0.15 | 65.7 | 49.2 | 60.3 | - |
| BLIP3-o (Chen et al., 2025a) | 8B | 0.84 | - | - | - | - | - | - | 79.7 | 52.8 | 68.0 | 81.60 |
| OmniGen2 (Wu et al., 2025b) | 7B | 0.80 | 1.00 | 0.95 | 0.64 | 0.88 | 0.55 | 0.76 | 79.6 | 52.4 | 70.5 | 83.57 |
| Bagel (Deng et al., 2025) | 14B | 0.82 | 0.99 | 0.94 | 0.81 | 0.88 | 0.64 | 0.63 | 66.2 | 37.5 | 43.9 | 85.07 |
| GPT-4o (OpenAI, 2024) | - | 0.85 | 0.99 | 0.92 | 0.85 | 0.91 | 0.75 | 0.66 | - | - | - | - |
| *AR-Only* | | | | | | | | | | | | |
| LLaMAGen (Sun et al., 2024) | 3.1B | 0.32 | 0.71 | 0.34 | 0.21 | 0.58 | 0.07 | 0.04 | - | - | - | 65.16 |
| VILA-U (Wu et al., 2024) | 7B | 0.40 | 0.88 | 0.42 | 0.25 | 0.69 | 0.08 | 0.09 | 56.8 | 43.3 | 50.1 | - |
| Emu3 (Wang et al., 2024) | 8B | 0.54 | 0.98 | 0.71 | 0.34 | 0.81 | 0.17 | 0.21 | 61.1 | 47.3 | 61.8 | 80.60 |
| Infinity (Han et al., 2024) | 8B | 0.73 | - | 0.85 | - | - | 0.49 | 0.57 | - | - | - | 83.46 |
| Janus-Pro-1B (Chen et al., 2025b) | 1B | 0.73 | 0.98 | 0.82 | 0.51 | 0.89 | 0.65 | 0.56 | 55.1 | 37.8 | 47.6 | 82.63 |
| Janus-Pro-7B (Chen et al., 2025b) | 7B | 0.80 | 0.99 | 0.89 | 0.59 | 0.90 | 0.79 | 0.66 | 63.6 | 35.3 | 49.4 | 84.17 |
| Lumina-mGPT 2.0 (Xin et al., 2025) | 7B | 0.80 | 1.00 | 0.92 | 0.57 | 0.88 | 0.70 | 0.72 | 58.5 | 36.8 | 47.2 | 79.11 |
| T2I-R1 (Jiang et al., 2025) | 7B | 0.79 | 0.99 | 0.91 | 0.53 | 0.91 | 0.76 | 0.65 | 81.3 | 58.5 | 72.4 | 84.42 |
| **Janus-FocusDiff-1B** | 1B | 0.82 | 0.99 | 0.93 | 0.59 | 0.90 | 0.80 | 0.68 | 61.5 | 47.7 | 60.4 | 83.17 |
| Janus-Pro-7B+SFT | 7B | 0.81 | 0.99 | 0.90 | 0.61 | 0.91 | 0.80 | 0.67 | 68.2 | 42.4 | 53.0 | 84.28 |
| **Janus-FocusDiff-7B** | 7B | 0.87 | 0.99 | **0.96** | 0.67 | **0.94** | **0.87** | 0.77 | 83.2 | 60.7 | 74.3 | 85.31 |

Janus-FocusDiff achieves better global vision-language alignment, with the average performance on PairComp outperforming the latest industry-leading models such as HunyuanImage-2.1, Omni-Gen2 and Bagel. Its performance ranks just below Qwen-Image, *which leverages training data and model parameters far exceeding ours*. Compared to the backbone model Janus-Pro-7B, the average values of $s_a$ and $s_g$ have achieved ***substantial improvements of 9.5 and 13.1 points***, respectively. Furthermore, when compared to T2I-R1 and Janus-Pro-R1, baseline models that similarly employ RL based on Janus-Pro, Janus-FocusDiff also demonstrates superior performance across all sub-tasks. **(2) Enhanced Generation Stability** is achieved with a significantly reduced gap between $s_a$ and $s_g$, with only an average 1.5-point difference. This gap matches that of Qwen-Image and is far smaller than the gap between $s_a$ and $s_g$ observed in other baseline models. This further demonstrates that our method achieves better fine-grained text-image semantic alignment, allowing the MLLM to focus on the subtle semantic differences in prompts for stable, high-quality image generation.

Moreover, to mitigate any potential reward hacking, we also **use Qwen2.5-VL-72B as the evaluator for PairComp**. As shown in Table 2, with the Qwen2.5-VL as the new evaluator, the relative performance ranking of models mostly remains unchanged. Janus-FocusDiff continues to outperform all baselines except Qwen-Image, with the average performance on PairComp outperforming the latest industry-leading models such as HunyuanImage-2.1, OmniGen2, and Bagel. Compared to the backbone model Janus-Pro-7B, the average values of $s_a$ and $s_g$ have achieved *substantial improvements of 11.2 and 15.8 points*, respectively. These results confirm that our evaluation protocol is unbiased and underscore the superiority of our model.

## 4.2 MAIN RESULTS ON EXISTING BENCHMARKS

We further conduct zero-shot evaluation on 3 text-to-image benchmarks: GenEval (Ghosh et al., 2023), T2I-CompBench (Huang et al., 2023), and DPG-Bench (Hu et al., 2024). The comparison against diffusion-only, AR-only, and hybrid methods is shown in Table 3. We have following observations:

**(1)** In most settings, our model achieves superior performance than various leading models. For example, on the GenEval benchmark, the overall performance of Janus-FocusDiff is even on par with that of GPT-4o. This underscores that we endow the MLLM with enhanced capability of vision-language alignment. **(2)** Compared to other baselines (*i.e.*, Show-o+PARM, T2I-R1) that also incorporate RL into AR-related text-to-image generation, our method achieves superior performance. For example, it consistently outperforms T2I-R1 on all of three benchmarks, highlighting the effectiveness of our pair-GRPO. **(3)** Compared to the backbone model Janus-Pro-7B, our method achieves performance improvements of 8.8% on Geneval and 55.2% on T2I-Compbench, significantly enhancing the capabilities of base model with strong effectiveness.

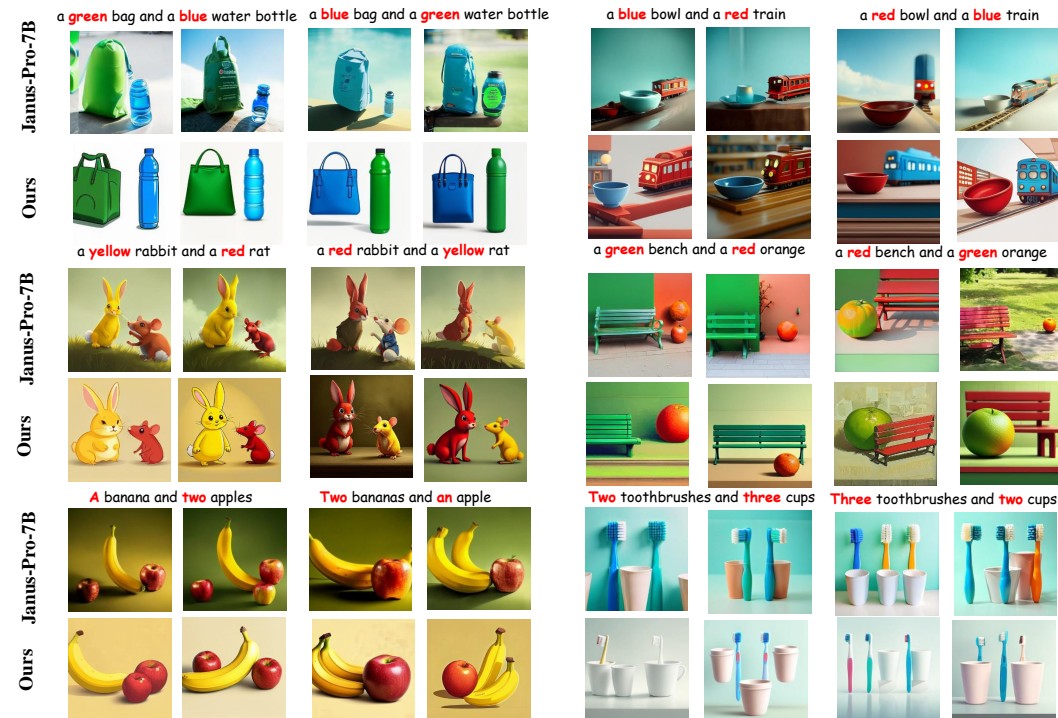

Figure 5: Qualitative Comparisons between Janus-Pro-7B and our Janus-FocusDiff-7B on pairs of similar prompts. For each prompt, we ask each model to generate two images.

### 4.3 QUALITATIVE EXAMPLES

**Image Generation with Similar Prompt Pairs.** Figure 5 and Figure 9 in Appendix present a direct qualitative comparison between Janus-FocusDiff-7B and Janus-Pro-7B on pairs of similar prompts with fine-grained semantic differences. For each prompt, we ask each model to generate two images. We can see that Janus-Pro-7B struggles to precisely control the fine-grained requirements of similar prompts. Moreover, even for the same prompt, the generated images are not consistently aligned with the target semantics. In contrast, our Janus-FocusDiff-7B is capable of accurately capturing the fine-grained semantic differences between prompts to generate corresponding images and stably produces high-quality images that meet the specified requirements.

**Image Generation with Counterfactual Prompts.** Endowing our model with fine-grained control over visual details, it can further generate images that more accurately match counterfactual prompts that are rarely found in the real world, as shown in Figure 6. For example, given the prompt "square watermelon", Janus-Pro-7B still generates a round one. In contrast, our Janus-FocusDiff successfully generates a watermelon with this counterfactual shape. This indicates that we effectively mitigate the issue of hallucination generation, eliminating the erroneous bias toward the training distribution.

### 4.4 IN-DEPTH ANALYSIS

Table 4: Ablation Study on GenEval &PairComp.

| | Methods | GenEval Overall↑ | PairComp-Overall | | PairComp-Avg | |
|---|---|---|---|---|---|---|
| | | | $s_a$ ↑ | $s_g$ ↑ | $s_a$ ↑ | $s_g$ ↑ |
| 1 | Janus-Pro-7B | 0.80 | 82.3 | 75.6 | 75.5 | 70.4 |
| 2 | Janus-FocusDiff-7B | **0.87** | **85.4** | **82.4** | **85.0** | **83.5** |
| 3 | w/o Group Expanding | 0.84 | 84.6 | 79.8 | 83.0 | 79.8 |
| 4 | w/o GT Image | 0.85 | 84.9 | 81.3 | 84.1 | 82.0 |
| 5 | Vanilla GRPO | 0.83 | 83.6 | 77.6 | 80.7 | 76.6 |
| 6 | w/o `FocusDiff-Data` | 0.82 | 82.7 | 76.0 | 77.0 | 71.9 |

**Effect of Individual Components.** To investigate the effectiveness of each component, we trained the following ablation models: **(1) w/o Group Expanding:** The group concept is restricted to images generated from a single prompt. **(2) w/o GT Image:** We set $p = 0.0$ and do not provide ground-truth images during RL. **(3) Vanilla GRPO:** We fully degrade Pair-GRPO to the vanilla GRPO. **(4) w/o `FocusDiff-Data`:** We select a set of commonly-used prompts to replace `FocusDiff-Data` for Vanilla GRPO training. As shown **Rows3-5** in Table 4, Pair-GRPO consistently outperforms other ablated algorithms on both Geneval and PairComp. Moreover, as shown in **Rows5-6**, the performance obtained from training with `Focusdiff-Data` outperforms that with commonly-used prompts. This indicates that both

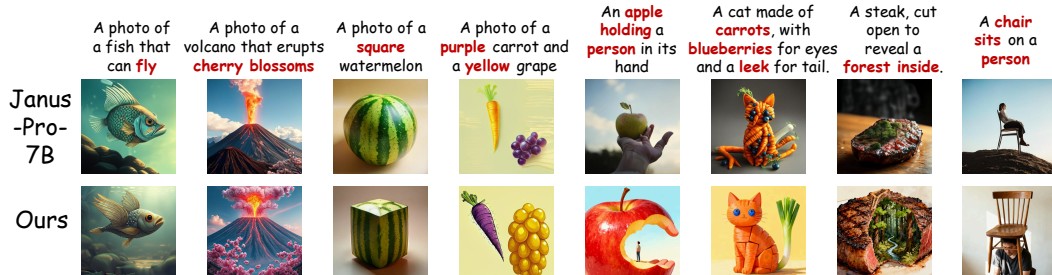

Figure 6: Image Generation with Counterfactual Prompts.

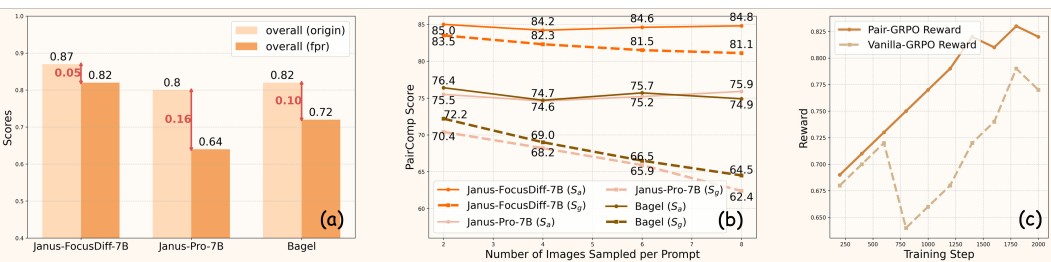

Figure 7: (a) Evaluation on GenEval with FPR overall score and original overall score. (b) Evaluation on PairComp when extending the number of images sampled per prompt from 2 to 8. (c) Reward trend for vanilla GRPO training and Pair-GRPO training.

Pair-GRPO and `FocusDiff-Data` enable the model to effectively focus on the fine-grained prompt requirements, thereby achieving better text-image alignment.

**Effect of Model Scale.** To further investigate the effectiveness of FocusDiff, we employ Janus-Pro-1B as the backbone and conduct training under the same settings. As shown in Tables 1 and 3, Janus-FocusDiff-1B demonstrates significant performance improvements compared to Janus-Pro-1B across all four benchmarks (*e.g.*, 20.7% on T2i-CompBench, 12.4% on PairComp), even outperforming Janus-Pro-7B on GenEval, which further validates the effectiveness of our approach.

**Stricter Evaluation on GenEval with FPR.** We introduce a stricter evaluation metric on GenEval termed Full-Pass Rate (FPR). FPR is defined as the success rate of prompts, where a prompt is deemed successful only if every image generated for that prompt is correct. As illustrated in Figure 7(a), both Janus-Pro-7B and Bagel exhibit a pronounced drop in FPR relative to the standard GenEval metric. In contrast, Janus-FocusDiff-7B not only attains a substantially higher FPR than these baselines, but also incurs a marginal decline of merely 5 points. These results demonstrate that our model achieves both a higher success rate and superior stability in image generation.

**Stricter Evaluation on PairComp with Extended Sampling.** We also conduct stricter evaluation on PairComp by gradually extending the number of images sampled per prompt from 2 to 8, with the resulting performance trends illustrated in Figure 7(b). For Janus-Pro and Bagel, increasing the number of sampled images leads to a significant decline in the geometric mean, indicating poor generation stability. In contrast, for Janus-FocusDiff-7B, the geometric mean decreases only very slightly as the number of generated samples increases. These confirm that our model consistently generates high-quality images with enhanced generation stability.

**Stability of Pair-GRPO Training** As acknowledged in (Yu et al., 2025; Xiong et al., 2025), GRPO is notoriously difficult to train due to its inherent instability. However, we find that Pair-GRPO not only surpasses vanilla GRPO in performance, but also enhances the stability and convergence of training. In Figure 7(c), we present the reward trends at different training steps for both vanilla GRPO and Pair-GRPO. Compared with the fluctuating rewards of vanilla GRPO, Pair-GRPO demonstrates a steady reward improvement, which suggests that it effectively optimizes the training instability.

**Choice of Evaluation Models.** For both PairComp evaluation and RL training, we choose InternVL2.5-26B as the evaluator as it reliably captures fine-grained semantic alignment between text and image. To verify this, we select 70 images rated by human annotators and leverage Qwen2.5VL-32B/72B, InternVL2.5-26B, LLaVA1.6-32B, GLM-4v-9B, CogVLM2-19B and GPT-4o for VLM evaluation. Then we report Pearson correlation to measure the alignment with human evaluation for each VLM. As shown in Table 5, InternVL2.5-26B and

Table 5: Pearson correlation with human evaluations for each VLM.

| Model | Pearson-r↑ |
|---|---|
| Qwen2.5VL-32B (Bai et al., 2025) | 0.72 |
| Qwen2.5VL-72B (Bai et al., 2025) | **0.79** |
| InternVL2.5-26B (Chen et al., 2024) | 0.77 |
| LLaVA1.6-32B (Liu et al., 2024c) | 0.68 |
| GLM4v-9B (GLM et al., 2024) | 0.64 |
| CogVLM2-19B (Hong et al., 2024) | 0.72 |
| GPT4 (Achiam et al., 2023) | 0.84 |

Qwen2.5-VL-72B achieve the highest correlation, with strongest ability to judge fine-grained image-text consistency among open-source models. So we select InternVL2.5-26B and Qwen2.5-VL-72B as our primary evaluator.

## 5 RELATED WORK

In recent years, diffusion models (Labs, 2024; Esser et al., 2024) have dominated the realm of visual generation. However, some efforts have explored using autoregressive (AR) models (Wang et al., 2024; Chen et al., 2025b) to generate images and achieved comparable performance. These approaches leverage an image tokenizer (Esser et al., 2021) to first encode images into discrete tokens, followed by a decoder that generates images through next-token prediction. Going one step further, hybrid models such as OmniGen2 (Wu et al., 2025b), Blip3-o (Chen et al., 2025a), Bagel (Deng et al., 2025) have emerged, integrating AR models with diffusion techniques to exploit both the strong reasoning capacity of AR and the high generation quality of diffusion. Furthermore, the AR property also satisfies the optimality condition of policy improvement, which supports effective RL post-training for visual generation (Guo et al., 2025b; Jiang et al., 2025), akin to the practice in LLMs (Guo et al., 2025a). Nevertheless, most existing methods focus primarily on overall semantics and struggle with fine-grained text–image alignment. In contrast, our FocusDiff enables AR-based models to exert precise control over visual tokens for stable and high-quality image generation.

## 6 CONCLUSION

In this paper, we propose **PairComp**, a new benchmark for text-to-image generation, revealing that existing models struggle with fine-grained text-image alignment. And we introduce **FocusDiff**, a training paradigm with a novel training dataset and an improved RL algorithm, enhancing fine-grained text-image semantic alignment by focusing on subtle differences between similar text-image pairs. On this basis, we develop Janus-FocusDiff, achieving superior performance on existing text-to-image benchmarks and significantly outperforming most prior prominent methods on PairComp.

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

APPENDIX OVERVIEW

The anonymous project of our paper is in `https://anonymous.4open.science/r/FocusDiff_Anonym-1F44`. In this supplementary material, we present:

- Vanilla GRPO for Autoregressive Image Generation in Section A.
- More Details on PairComp in Section B.
- More Details on `FocusDiff-Data` in Section C.
- Implementation Details in Section D.
- Evaluation Details in Section E.
- More Experimental Results in Section F.
- The Use of Large Language Models in Section G.
- Ethics Statement and Reproducibility Statement in Section H.

## A   VANILLA GRPO FOR AUTOREGRESSIVE IMAGE GENERATION.

We adopt Group Relative Policy Optimization (GRPO) as the framework for reinforcement learning, GRPO enhances PPO by eliminating the value function and estimating the advantages in a group-relative manner. Specifically, given the input prompt $\mathcal{T}$, the old policy $\pi_{\theta_{old}}$ first samples a group of $G$ individual images as the response group $\mathcal{G} = \{\mathcal{I}_i^1\}_{i=1}^G$. We input each response with the group into the reward function to obtain the individual reward $\mathtt{R}_{\mathcal{I}_i}$. We then calculate the advantages $\{A_i\}_{i=1}^G$, where each $A_i$ measures the relative quality of output compared to the average reward:

$$A_i = \frac{\mathtt{R}_{\mathcal{I}_i} - \mathrm{mean}\big(\{\mathtt{R}_{\mathcal{I}_i}\}_{i=1}^G\big)}{\mathrm{std}\big(\{\mathtt{R}_{\mathcal{I}_i}\}_{i=1}^G\big)} \tag{2}$$

Then, we update the policy network parameters by the following training loss:

$$\mathcal{J}(\theta) = \mathbb{E}_{\substack{(\mathcal{T},a)\sim\mathcal{D} \\ \{y_i\}_{i=1}^G \sim \pi_{\theta_{old}}(\cdot|\mathcal{T})}} \left[ \frac{1}{\sum_{i=1}^G |y_i|} \sum_{i=1}^G \sum_{j=1}^{|y_i|} \left( \min\big(\rho_{i,j}A_i, \mathrm{clip}\big(\rho_{i,j}, 1-\varepsilon, 1+\varepsilon\big)A_i\big) - \beta D_{\mathrm{KL}} \right) \right], \tag{3}$$

where $D_{\mathrm{KL}} = \frac{\pi_{ref}}{\pi} - \log\frac{\pi_{ref}}{\pi} - 1$ is the the KL divergence to maintain training stability. And $\rho_{i,j} = \frac{\pi_\theta(y_{i,j}|\mathcal{T},y_{i,<j})}{\pi_{\theta_{old}}(y_{i,j}|\mathcal{T},y_{i,<j})}$ is the ratio between probabilities of $\pi_\theta$ and $\pi_{\theta_{old}}$ for outputting current token.

## B   MORE DETAILS ON PAIRCOMP

Each test case in PairComp contains two similar prompts with subtle differences. The two prompts exhibit word-level differences that lead to noticeable distinctions in six types of fine-grained semantic aspects: (1) Overall appearance difference; (2) Color difference; (3) Counting difference; (4) Position difference; (5) Style & Tone difference; (6) Text difference. Next, we will provide a detailed explanation of these six types.

- **Color:** highlighting differences in the color of specific items in two images. For example, an umbrella in one picture is purple while in another picture it is green.
- **Position:** Differences in the relative positioning of specific items in two images. For example, in one picture object `[A]` is to the left of object `[B]` while in another picture `[A]` is to the right of `[B]`.
- **Text:** Differences in the textual content on an item in two images. For example, the departure time on a ticket in one picture is "20:00" while the departure time on a ticket in another picture is "21:00".
- **Style & Tone:** The differences can be categorized into two types: (1) Differences in the overall style of two images. For example, one picture is in an oil painting style while another picture is in an ink wash painting style. (2) Differences in the overall atmosphere (weather, season, etc.) in two images. For example, the scene depicted in one picture is on a sunny day while the scene depicted in another picture is on a foggy day.

- **Counting:** Differences in the quantity of specific items in two images. For example, there are 3 eggs in one picture while there are only 2 eggs in another picture.

- **Overall-appearance:** Differences in the overall appearance of items in two images, including but not limited to the previously mentioned item such as color, as well as previously unmentioned decorations or style differences of objects. For example, a cat in one picture is wearing a bow tie while a cat in another picture is wearing a bell.

## C    MORE DETAILS ON FOCUSDIFF-DATA

In this section, we give more details on how to construct `FocusDiff-Data` from the image editing dataset (Zhao et al., 2024a; Yu et al., 2024), with the pipeline shown in Figure 8. In the first step, considering the potential poor quality of the image editing dataset, we conduct data cleaning on the raw data to retain only high-quality samples. Using the InternVL2.5-26B model, providing it with the before-after-editing images and the editing instruction, we evaluate three key aspects with the following prompts: *(1) whether the edited image follows the editing instructions*; *(2) whether the non-edited areas of the edited image remain consistent with the original image*; and *(3) whether the overall quality and natural appearance of the edited image are acceptable*. We filter out any pairs that fail to meet these criteria.

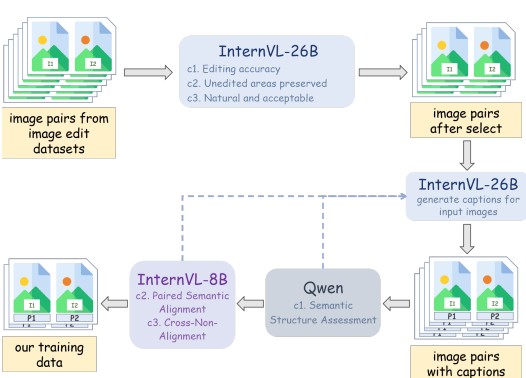

Figure 8: The pipeline for constructing `FocusDiff-Data`

Subsequently, we input the pair of before-and-after images along with the editing instructions into InternVL2.5-26B (Chen et al., 2024). We prompt it to generate a pair of captions for the images that share a similar stylistic structure but differ only in individual words, thereby highlighting the differences between the images. The task prompt is formatted as:

> The user will provide an original image and an edited image based on specific editing instructions. Your task is to write a description for each of these two images. The descriptions must adhere to the following guidelines.
> **Identical Structure:** Both descriptions should follow the exact same structural format. Ensure that verbs, adjectives, and other parts of speech align in number and position between the two sentences.
> **Minimal Differences:** Only one to three words should be altered between the original and edited image descriptions to emphasize the changes made.
> **Direct Comparison:** The paired sentences should correspond directly, allowing for a clear comparison between the original and edited images.

After generating the captions $(\mathcal{P}_1, \mathcal{P}_2)$ for the images $(\mathcal{I}_1, \mathcal{I}_2)$, we conduct a post-verification operation with three conditions: **(1)** Using the Qwen model (Bai et al., 2023), we assess whether $\mathcal{P}_1$ and $\mathcal{P}_2$ exhibit similar semantic structures; **(2)** Using the InternVL-8B model (Chen et al., 2024), we verify whether $\mathcal{P}_1$ and $\mathcal{I}_1$, as well as $\mathcal{P}_2$ and $\mathcal{I}_2$, were semantically aligned. **(3)** We further leverage InternVL-8B to ensure that $\mathcal{P}_1$ and $\mathcal{I}_2$, as well as $\mathcal{P}_2$ and $\mathcal{I}_1$, are not semantically aligned. If all of three conditions are satisfied, the sample is deemed valid and included in our training dataset. Otherwise, we request the InternVL2.5-26B model to regenerate captions for the two images and conduct the post-verification again. If the post-verification still fails, the image pair is then discarded. Finally, we retained approximately $200,000$ high-quality data pairs.

## D IMPLEMENTATION DETAILS

**Supervised Fine-Tuning.** We first leverage FocusDiff-Data to conduct autoregressive text-to-image supervised fine-tuning on Janus-Pro. The objective function is $p(y) = \frac{1}{S} \sum_{i=1}^{S} \log P_\theta(y_i|y_{<i}, \mathcal{T})$, where $y$ is the visual token of an image with $S$ as the sequence length, $\mathcal{T}$ is the text condition. The detailed hyperparameters for training are shown in Table 6.

**Reward Calculation.** The overall design philosophy of our reward model is to leverage QA-based visual comprehension (Chen et al., 2024; Liu et al., 2024b; Bai et al., 2025) models, which will return a consistency score $R^{QA}(\cdot) \in [0,1]$ for each text-image pair. We leverage InternVL2.5-26B (Chen et al., 2024) as the reward model to provide appropriate incentives, which will return a consistency score $R_\mathcal{I} \in [0,1]$ for each text-image pair. Specifically, for short prompts, we directly the MLLM with the question "Does this image match the description? Please directly respond with yes or no". We record the probability of the model responding with "Yes" as $P_{yes}$ and "No" as $P_{no}$, with the reward score calculated as $S(\mathcal{I}, \mathcal{P}) = P_{yes}/(P_{yes} + P_{no})$. For long prompts, inspired by , we first decompose the prompt to semantic tuples (*e.g.*, attribute, and spatial relation) and then generate yes-or-no questions (*e.g.*, "Is the cdog red?"). The MLLMs are asked to perform a VQA task for the prompt and generated image, returning a score of 0 to 1 for each question in the same way. The reward is obtained by averaging the evaluation of the MLLMs on multiple questions for a prompt.

**Reinforcement Learning.** Our proposed Pair-GRPO is an improved version of GRPO, with training prompts sourced from FocusDiff-Data. We set the $G = 7$, first expanding the group size from 7 to 14. There is a probability $p$ that the group size may further increase to 18, as we introduce ground-truth images corresponding to prompt pairs from `FocusDiff-Data` and pair them with the prompts. The probability $p$ is dynamic, decreasing from 1.0 at the start of training to 0.0 by the end.

During RL training, we used the fine-tuned Janus-Pro as the backbone model and set the

Table 6: The detailed training hyper-parameters of supervised fine-tuning and reinforcement learning.

| Hyper-parameters | Fine-Tuning | Reinforcement Learning |
|---|---|---|
| Optimizer | AdamW | AdamW |
| Optimizer param. | $\beta_1 = 0.9, \beta_2 = 0.95, \epsilon = 1e{-}6$ | |
| Peak LR | 2.0e-5 | 1.0e-5 |
| Convert LR | - | 2.0e-6 |
| Convert step | - | 300 |
| Min LR | 2.0e-7 | 2.0e-7 |
| LR scheduler | Cosine | Linear+Cosine |
| Batch size | 256 | 128 |
| Training Steps | 5K | 2.2K |
| Warmup Steps | 100 | 100 |
| Weight decay | 0.05 | 0.05 |
| Gradient clipping | 1.0 | 1.0 |
| Numerical precision | bfloat16 | bfloat16 |
| Resource Usage | 8 NVIDIA A800 | 16 NVIDIA A800 |

batch size to 128, meaning that each optimization iteration includes 128 different prompts. All parameters are tunable. We totally conduct 2200 iterations of post-training optimization. We find that the learning rate is crucial: a learning rate that is too small results in insignificant performance gains, while a learning rate that is too large leads to unstable training. To address this, we design a combined Linear + Cosine learning rate scheduler. The learning rate quickly drops linearly from a peak value to a lower "convert learning rate" at a "convert step", and then gradually decreases along a cosine curve. However, we still encounter some instability during training, indicated by a downward trend in the reward curve. To address this, we adopt the following measures: (1) When the reward curve dropped sharply, we reduce the learning rate to half or two-thirds of its current value and resume the training; (2) When the reward curve declined gradually, it suggests that the KL divergence constraint with a less capable reference model is limiting the model improvement. Thus we update the reference model to the current model and then resume the training. The detailed hyperparameters for training are shown in Table 6.

**Inference.** During inference, we follow the inference setup of Janus-Pro, setting $topk = 4096$ for visual token sampling. Besides, we use classifier-free guidance on the logits for autoregressive sampling in a manner similar to (Wang et al., 2024; Liu et al., 2024a). We set the guidance scale to 4.0, or 5.0 or 6.0.

## E EVALUATION DETAILS

**Baseline.** We compare Janus-FocusDiff against three primary categories of models: diffusion-only methods, autoregressive (AR)-only methods, and hybrid approaches that combine both diffusion and

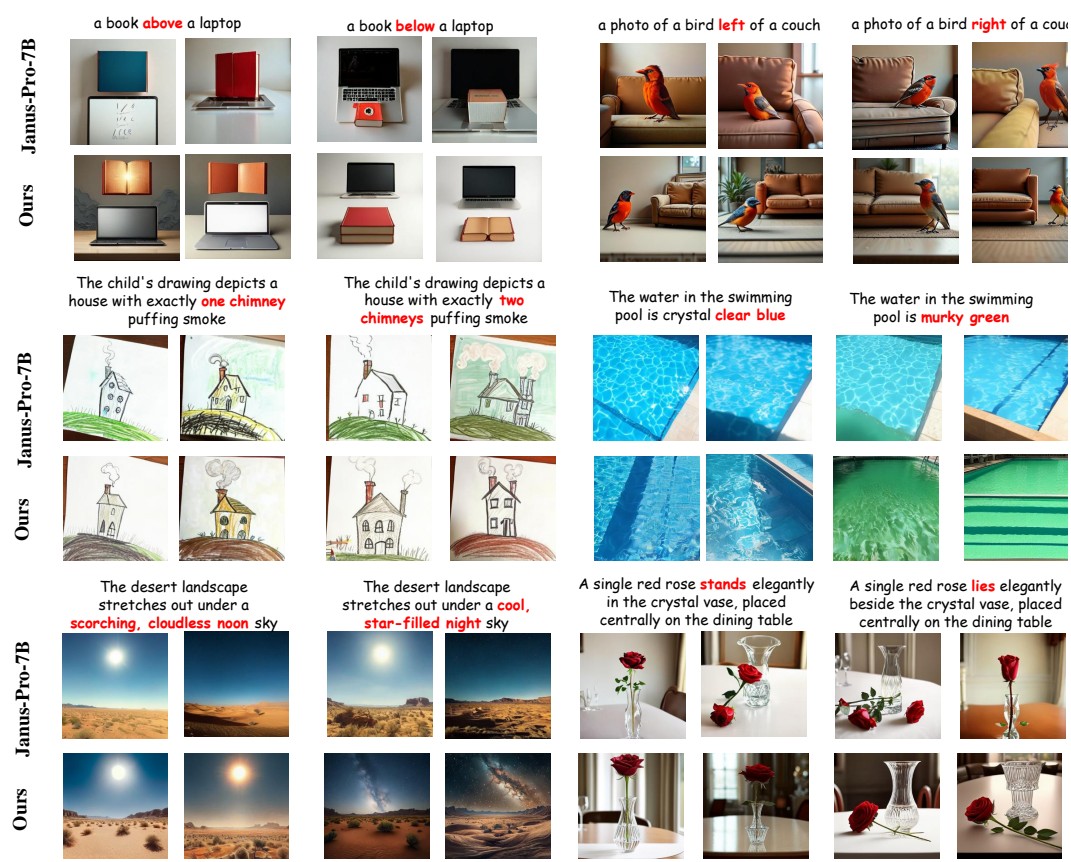

Figure 9: More qualitative Comparisons between Janus-Pro-7B and Janus-FocusDiff-7B on pairs of similar prompts.

AR techniques. The diffusion-only baselines include SD3 (Esser et al., 2024), FLUX.1-dev (Labs, 2024), Sana-1.5 (Xie et al., 2025), Janus-Flow (Ma et al., 2024), Lumina-Image2.0 (Qin et al., 2025), HiDream-I1-Full (Cai et al., 2025), HunyuanImage-2.1 (Team, 2025), Qwen-Image (Wu et al., 2025a). The AR-only baselines include LLamaGen (Sun et al., 2024), VILA-U (Wu et al., 2024), Emu3 (Wang et al., 2024), Infinity (Han et al., 2024), Janus-Pro-1B (Chen et al., 2025b), Janus-Pro-7B (Chen et al., 2025b), Lumina-mGPT 2.0 (Xin et al., 2025), T2I-R1 (Jiang et al., 2025), Janus-Pro-R1 (Pan et al., 2025). The hybrid baselines include Show-o (Xie et al., 2024), Show-o+PARM (Guo et al., 2025b), SEED-X (Ge et al., 2024), BLIP3-o (Chen et al., 2025a), Omni-Gen2 (Wu et al., 2025b), Bagel-Think (Deng et al., 2025), X-Omni-En (Geng et al., 2025), GPT-4o (OpenAI, 2024), . It is worth noting that, Show-o+PARM, T2I-R1 and Janus-Pro-R1 attempt to enhance the text-to-image generation capabilities of AR-based MLLMs through reinforcement learning.

**Benchmarks.** In PairComp, we leverage InternVL2.5-26B as the main evaluation model (Qwen2.5-72B as the assistant evaluation model) with the prompt: "Does this image match the description? Please directly respond with yes or no." We record the probability of the model responding with "yes" (denoted $P_{yes}$) and with "no" (denoted $P_{no}$), with the semantic consistency score calculated as $S(\mathcal{I}, \mathcal{T}) = P_{yes}/(P_{yes} + P_{no})$. For each prompt, we require a text-to-image model to generate two images. Therefore, for a pair of similar prompts $(\mathcal{T}_i^1, \mathcal{T}_i^2)$, we obtain four generated images $(\mathcal{I}_i^{1,1}, \mathcal{T}_i^{1,2}, \mathcal{I}_i^{2,1}, \mathcal{I}_i^{2,2})$. We then compute the semantic consistency scores for each image with respect to its corresponding prompt: $s_i^{1,1} = S(\mathcal{I}_i^{1,1}, \mathcal{T}_i^1)$, $s_i^{1,2} = S(\mathcal{I}_i^{1,2}, \mathcal{T}_i^1)$, $s_i^{2,1} = S(\mathcal{I}_i^{2,1}, \mathcal{T}_i^2)$, $s_i^{2,2} = S(\mathcal{I}_i^{2,2}, \mathcal{T}_i^2)$. The arithmetic mean score is calculated as: $s_a = \frac{1}{4N} \sum_{i=1}^{N} (s_i^{1,1} + s_i^{1,2} + s_i^{2,1} + s_i^{2,2})$, and the geometric mean score is calculated as: $s_g = \frac{1}{N} \sqrt[4]{s_i^{1,1} \cdot s_i^{1,2} \cdot s_i^{2,1} \cdot s_i^{2,2}}$. The score of the geometric (arithmetic) mean for "Average" is obtained by averaging the geometric (arithmetic) mean scores of the other six sub-tasks.

Furthermore, we also conduct zero-shot evaluation on 3 existing text-to-image benchmarks: GenEval (Ghosh et al., 2023), T2I-CompBench (Huang et al., 2023), and DPG-Bench (Hu et al., 2024). GenEval contains 6 different subtasks of varying difficulty requiring various compositional skills, including `single object` (SingObj), `single object` (TwoObj), `counting`, `colors`, `position`, `color binding` (ColorAttri). And we adopt the metric proposed by (Ghosh et al., 2023) for evaluation. Each subtask is scored independently, and the overall score is calculated as the average of all six subtask scores. The T2I-CompBench encompasses three subtasks following Wang et al. (2024): `color`, `shape`, `texture`. Building on prior research, we employ the Blip-VQA score (Li et al., 2022) as the evaluation metric. While for DPG-Bench, we follow the metrics proposed in (Hu et al., 2024) to conduct evaluation.

## F    MORE EXPERIMENTAL RESULTS

In Figure 9 we present more qualitative comparison between Janus-FocusDiff-7B and Janus-Pro-7B on pairs of similar prompts with fine-grained semantic differences. Janus-Pro-7B struggles to precisely control the fine-grained requirements of similar prompts. Moreover, even for the same prompt, the generated images are not consistently aligned with the target semantics. In contrast, our Janus-FocusDiff-7B is capable of stably generating high-quality images that meet the specified requirements.

## G    THE USE OF LARGE LANGUAGE MODELS

Our use of large language models (LLMs) spans two main aspects. First, during the experimental phase, as described in Appendix C, we leveraged LLMs (InternVL) to assist in generating the high-quality `FocusDiff-Data` dataset, including image captioning and filtering. Second, during paper writing, we utilized GPT-4 to polish certain expressions within the manuscript, enhancing the clarity and fluency of the text.

## H    ETHICS STATEMENT AND REPRODUCIBILITY STATEMENT

**Ethics Statement.**    This study does not raise any ethical concerns. The research does not involve subjective assessments or the use of private data. All experiments draw exclusively on publicly available datasets or data generated by open-source multimodal large language models.

**Reproducibility Statement.**    The anonymous repository for our paper is available at `https://anonymous.4open.science/r/FocusDiff_Anonym-1F44`. In this anonymous repository, we provide the complete training codebase, the PairComp benchmark for evaluating fine-grained text-image alignment, and the corresponding inference scripts for better reproducibility.

Table 7: Performance comparison between pointwise reward and pairwise reward.

| Methods | GenEval | DPG-Bench | PairComp (InternVL2.5) | | PairComp (Qwen2.5-VL) | |
|---|---|---|---|---|---|---|
| | Overall↑ | Avg↑ | Avg-$S_a$↑ | Avg-$S_g$↑ | Avg-$S_a$↑ | Avg-$S_g$↑ |
| Janus-Pro-7B | 0.80 | 84.17 | 75.5 | 70.4 | 63.6 | 54.1 |
| Janus-FocusDiff (Pairwise Reward) | 0.84 | 84.90 | 82.0 | 79.8 | 71.3 | 65.6 |
| Janus-FocusDiff (Pointwise Reward) | **0.87** | **85.31** | **85.0** | **83.5** | **74.8** | **69.9** |

Table 8: Evaluation results on PairComp based on InternVL2.5-26B, when decomposing the prompts into sub-questions for evaluation.

| Methods | Overall Appear. | | Color | | Counting | | Position | | Style&Tone | | Text | | Average | |
|---|---|---|---|---|---|---|---|---|---|---|---|---|---|---|
| | $S_a$↑ | $S_g$↑ | $S_a$↑ | $S_g$↑ | $S_a$↑ | $S_g$↑ | $S_a$↑ | $S_g$↑ | $S_a$↑ | $S_g$↑ | $S_a$↑ | $S_g$↑ | $S_a$↑ | $S_g$↑ |
| LLamaGen (Sun et al., 2024) | 60.8 | 56.3 | 69.4 | 64.4 | 48.1 | 42.1 | 50.4 | 43.7 | 75.5 | 66.8 | 27.4 | 20.9 | 55.3 | 49.0 |
| Show-o (Xie et al., 2024) | 75.9 | 72.6 | 89.1 | 87.0 | 62.2 | 58.9 | 52.0 | 47.4 | 92.4 | 89.3 | 42.1 | 34.2 | 68.9 | 64.9 |
| Bagel-Think (Deng et al., 2025) | 82.3 | 79.8 | 92.1 | 90.6 | 72.7 | 69.8 | 72.7 | 67.5 | 91.1 | 87.4 | 65.1 | 58.4 | 79.3 | 75.6 |
| Janus-Pro-7B (Chen et al., 2025b) | 84.1 | 81.3 | 96.4 | 94.8 | 55.4 | 49.2 | 72.6 | 68.2 | 94.5 | 91.2 | 65.4 | 58.5 | 78.1 | 73.9 |
| SD3 (Esser et al., 2024) | 87.0 | 85.2 | 96.1 | 95.4 | **78.6** | **75.8** | 74.8 | 71.6 | 90.5 | 87.3 | **95.0** | **94.1** | 87.0 | 84.9 |
| Janus-FocusDiff-7B | **89.3** | **87.9** | **98.6** | **98.3** | 72.0 | 70.1 | **78.1** | **75.2** | **95.4** | **94.9** | 88.1 | 86.8 | **86.9** | **85.5** |

Table 9: Performance Comparison on all six subtasks of T2I-CompBench.

| Methods | Color↑ | Shape↑ | Texture↑ | Spatial↑ | Non-Spatial↑ | Complex↑ |
|---|---|---|---|---|---|---|
| FLUX.1-dev (Labs, 2024) | 0.74 | 0.57 | 0.69 | 0.29 | 0.31 | 0.37 |
| Show-o (Xie et al., 2024) | 0.56 | 0.41 | 0.46 | 0.20 | 0.30 | 0.29 |
| Show-o+PARM (Guo et al., 2025b) | 0.75 | 0.56 | 0.66 | 0.29 | 0.31 | 0.37 |
| Janus-Pro-7B (Chen et al., 2025b) | 0.64 | 0.35 | 0.49 | 0.21 | 0.31 | 0.36 |
| T2I-R1 (Jiang et al., 2025) | 0.81 | 0.59 | 0.72 | 0.34 | 0.31 | 0.40 |
| Janus-FocusDiff-7B | **0.83** | **0.61** | **0.74** | **0.35** | **0.33** | **0.42** |

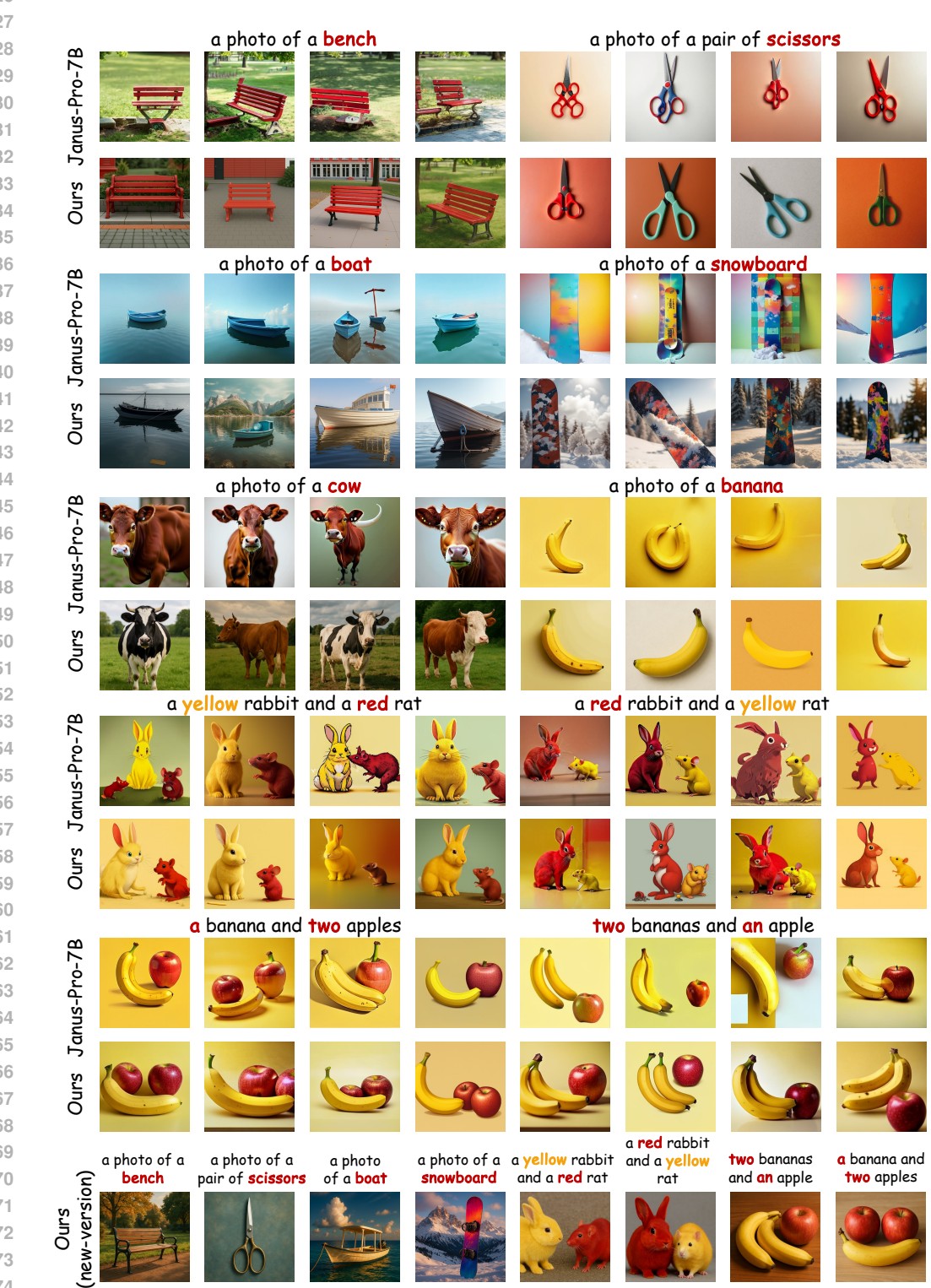

Figure 10: More qualitative Comparisons between the image generated by Janus-Pro-7B and Janus-FocusDiff-7B on various prompts. The images generated by our model generally achieve better aesthetic quality. **In the last row of this figure, we also show images generated by the new version of Janus-FocusDiff, which is trained by an improved version of FocusDiff-Data.**

A **dog** chases a **girl**.  A **girl** chases a **dog**.

A **dog** is to the left of a **pig**.

A **pig** is to the left of a **dog**.

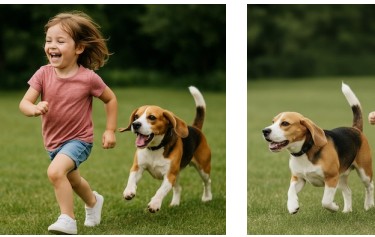 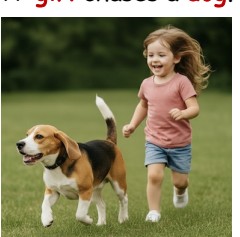 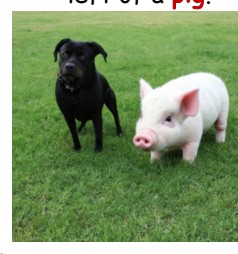 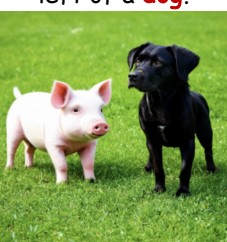

(a) Data examples in EvoGen

The bow of a cargo vessel rises above the sea under overcast light, with a seagull **standing**/**flying** near it.

A person wearing a dark blue top and jeans walks down the street with a dollar bill in his **pocket**/**hand**.

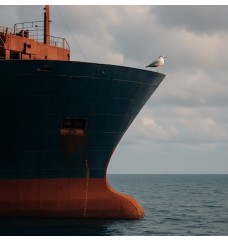 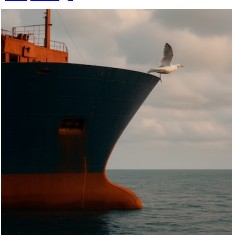 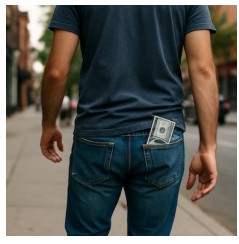 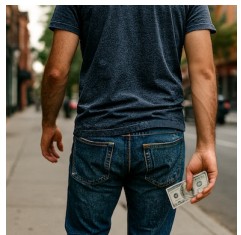

A table filled with food, such as a pepperoni pizza, a cheeseburger with vegetables, fries with a **generous**/**small** amount of ketchup, and fried bites served in a bowl.

A desk with a desktop computer, a **round**/**square** coffee cup, and yellow and pink sticky notes.

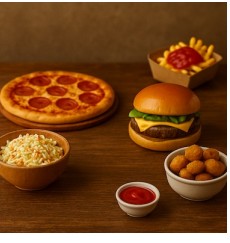 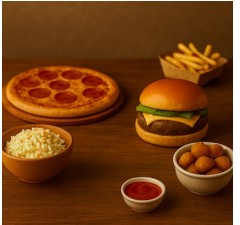 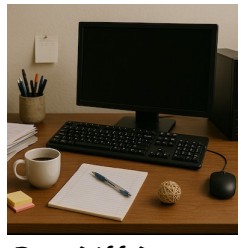 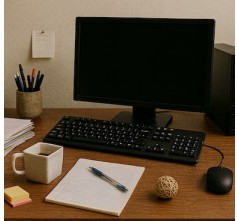

(b) Data examples in FocusDiff-Data

Figure 11: We present some typical datat examples in EvoGen and FocusDiff-Data. Compared to EvoGen, FocusDiff-Data not only builds upon compositional learning of relational terms, but also emphasizes fine-grained semantic details that are often overlooked beyond a prompt's primary semantics.

A **wooden**/**metal** chair beside the window.

A bicycle parked on the street at **sunset**/**noon**.

A tent in the **desert**/**snow**.

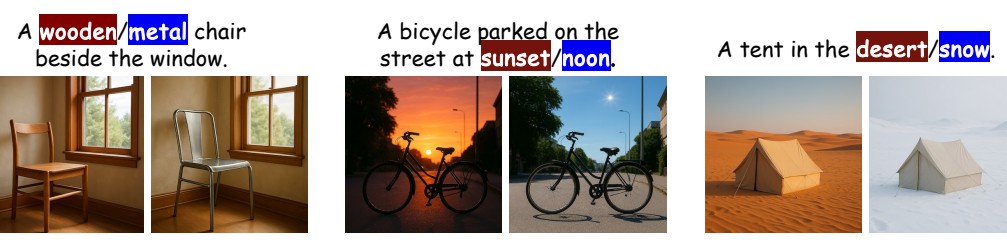

Figure 12: Data examples in high-aesthetic-quality editing datasets.

