# OpenReview forum: "FocusDiff: Advancing Fine-Grained Text-Image Alignment for Autoregressive Visual Generation through RL"
_ICLR.cc/2026/Conference — Submitted to ICLR 2026_

### Official Review · Reviewer_YYr2 · 2025-10-24

**Soundness:** 3
**Presentation:** 2
**Contribution:** 2
**Rating:** 6
**Confidence:** 3

**Summary:**

The paper introduces FocusDiff, a method improving fine-grained text-to-image alignment. Using the new PairComp benchmark, it shows current models fail to capture subtle semantic differences. FocusDiff employs a paired text-image dataset and a refined GRPO-based algorithm to focus on nuanced variations, achieving better semantic precision and outperforming prior methods on standard and PairComp benchmarks. However, there are some typos in this paper, I hope the authors can fix them.

**Strengths:**

[1]. The codes are provided in the supplementary materials, which helps researchers reproduce the results and makes the paper easier to understand.

[2]. This paper introduces PairComp and evaluates paired prompts that differ only in subtle word-level semantics.

[3]. The paper proposes Pair-GRPO, an extension of Group Relative Policy Optimization designed for fine-grained visual generation.

[4]. The experiments are comprehensive and effectively demonstrate the validity of the proposed method.

[5]. The paper have provided the webpage to help me understand.

**Weaknesses:**

[1]. The paper lacks sufficient novelty. The proposed method, GRPO, has already been widely adopted and explored in the field of large language models.

[2]. In 771 line, there is a typo. The phrase “with the pipeline shown in 8” appears to be incomplete or unclear.

[3]. The quality of the generated results is not satisfactory. The images or outputs appear neither aesthetically pleasing nor natural, espically in Figure 5, the results looks fall behind the diffusion model.

[4]. In Figure 3, the shape of the sign and car is changed. The authors should provide an explanation or analysis of this issue.

**Questions:**

See Weaknesses.

---

> ### Author Response · Authors · 2025-11-23
> **Response to Reviewer YYr2 Part 1/2**
>
> Thank you for your constructive feedback. We will address your concerns point by point.
>
> > **Weakness1:** The paper lacks sufficient novelty. The proposed method, GRPO, has already been widely adopted and explored in the field of large language models.
>
> **A1:** Thank you for the question. First, we want to clarify that the novelty of our paper does not lie in the introduction of GRPO for image generation, but **in our motivation and high-level design principles**:
>
> The **motivation** stems from the finding that existing AR-based image generation models struggle with fine-grained text-image alignment, consequently failing to realize precise control over visual tokens on detailed semantics. Based on this motivation, our **high-level design principle** is to transition from aligning a single text-image pair for vision-language global alignment to learning the subtle differences between two similar text-image pairs for fine-grained vision-language alignment. **This naturally leads to our greatest contribution**: *the innovative data construction pipeline and PairComp benchmark, the respective contributions of which are acknowledged by reviewers TGyE and 8cu7.*
>
> Moreover, we acknowledge that our Pair-GRPO algorithm is not a brand-new reinforcement learning algorithm but an improvement of GRPO, primarily through expanding the group concept and shifting the focus from exploitation to exploration. However, we argue that given the key contributions in design principles and data construction, the algorithmic refinement should align with the core motivation for practical improvement rather than solely pursuing innovation. And **Pair-GRPO naturally emerges from our motivation, closely aligning with our high-level design principles, and proving effective for solving fine-grained alignment.**
>
> &nbsp;
>
> > **Weakness2: In line 771, there is a typo. The phrase "with the pipeline shown in 8" appears to be incomplete or unclear.**
>
>
> **A2:** Thank you for pointing out the error. We intended to refer to the pipeline shown in Figure 8. In the new version of our paper, we have corrected this typo accordingly.
>
>
> &nbsp;
>
> > **Weakness3: The quality of the generated results is not satisfactory. The images or outputs appear neither aesthetically pleasing nor natural, especially in Figure 5, the results look fall behind the diffusion model.**
>
> **A3:** Thank you for your question. First, we want to clarify that our backbone model, Janus-Pro, is an autoregressive model for image generation, not a diffusion model. Compared to diffusion models, **images generated by Janus-Pro generally suffer from a significant aesthetic quality disadvantage**, frequently exhibiting noticeable artifacts and unnatural structures, such as the generation of distorted shapes (e.g., a misshapen bench). **While these defects are significantly improved in those generated by Janus-Pro-FocusDiff**. We have included a qualitative comparison in ***Appendix Figure 10 of the new version of our paper***, showcasing multiple generated images from the same prompt using both Janus-Pro and Janus-Pro-FocusDiff to further illustrate their respective aesthetic effects. **It indicates that the images generated by our model generally achieve better aesthetic quality than the backbone model.**
>
> However, as you observed, many cases from our model show an over-smoothing effect, a lack of texture and realism, resulting in a tendency to generate images with a painting-like style. **We attribute this sub-optimality to two main factors**. **Firstly**, the inherent aesthetic limitations of our backbone model contribute to the reduced quality.
>
> **Secondly**, and more critically, our training data **source from some earlier image editing datasets such as Anyedit and Ultraedit** (we conduct fine-tuning on these data before RL). Due to the suboptimal editing pipelines used to construct these datasets, **the images within them often suffer from artifacts like a loss of texture detail and realism**. Consequently, training on such data subsequently leads to the aesthetic and realism issues observed in our generated images.
>
> Moreover, over the past few months, some high-aesthetic-quality editing datasets, such as GPT-Image-1.5M and nanobanana-130K, have emerged. We have **continued to iterate and refine our FocusDiff-Data using these superior datasets**. This new version of FocusDiff-Data was then used to perform supervised fine-tuning and reinforcement learning on the Janus-Pro model. Consequently, **the resulting new version of our model substantially resolves the aesthetic issue**, as shown in ***Appendix Figure 10 of the new version of our paper***.

---

> > ### Author Response · Authors · 2025-11-23
> > **Response to Reviewer YYr2 Part 2/2**
> >
> > > **Weaknesses4: In Figure 3, the shape of the sign and car is changed. The authors should provide an explanation or analysis of this issue.**
> >
> > **A4:** Figure 3 in our paper illustrates the FocusDiff-Data training set. Crucially, the image pairs displayed in the figure are sourced directly from existing image editing datasets, rather than being generated by our model. Our primary contribution to these pre-existing image pairs is the generation of the corresponding text captions.
> >
> >
> > As stated in **A3**, these images originate from earlier image editing datasets like AnyEdit and UltraEdit, which have a suboptimal construction pipeline for post-edit image generation. Despite the sophisticated quality screening, it is still impossible to avoid the fact that the pre- and post-edited images selected from these datasets remain perfectly consistent in the unedited regions. Meanwhile, the post-edited image may also exhibit some minor deformations. In short, the shape change you observed is a **common issue inherent in current image editing training datasets**, **not a problem with the quality of our model's generation.**
> >
> >
> > Fortunately, in these months, high-aesthetic-quality editing datasets, such as GPT-Image-1.5M and nanobanana-130K, have emerged, which utilize GPT-4o or NanoBanana to annotate editing data, largely ensuring that the unedited regions remain perfectly preserved before and after editing, ***as shown in Appendix Figure 12 of the new version of our paper***. Consequently, the new version of FocusDiff-Data based on these high-aesthetic-quality image editing datasets largely alleviates the issue you mentioned.

---

> ### Comment · Reviewer_YYr2 · 2025-11-26
>
> Thanks for your rebuttal.
>
> I understand that your backbone is Janus-Pro rather than a diffusion model, and I hold different expectations for different types of models. My main concern is that even the figures presented in your paper fail to demonstrate decent visual results. Given this, how can you convince me of the effectiveness of your method?

---

> > ### Author Response · Authors · 2025-11-26
> >
> > Thank you for your concern. We would like to provide further clarification on three main points:
> >
> > **(1) The Origin of Aesthetic Issues**
> >
> > First, we acknowledge that our generated images indeed have considerable room for aesthetic improvement. Specifically, many cases show an over-smoothing effect, resulting in a tendency to generate images with a painting-like style, lacking texture detail and realism. A primary reason for this is that our training data is sourced from **earlier image editing datasets that lack high aesthetic quality**, such as Anyedit and Ultraedit, which utilize suboptimal editing pipelines for image construction. Consequently, the **images within these datasets inherently suffer from such aesthetic artifacts**, and subsequent SFT on these images naturally leads to the aesthetic and realism issues observed in our generated images.
> >
> > &nbsp;
> >
> > **(2) Core Research Focus (Semantic Consistency)**
> >
> > However, in our previously submitted version, the **core focus of our work is on image-text semantic consistency**. Our primary research objective is to demonstrate that the proposed FocusDiff-Data and training paradigm could **effectively achieve a significant improvement in fine-grained semantic alignment** within the generated images compared to the Janus-Pro backbone model. Conversely, we **do not specifically pay excessive attention to the aesthetic aspect**; we merely ensure that the generated images do not exhibit the obvious unnatural shape distortions that are extensively present in images generated by Janus-Pro.
> >
> > Based on this research focus, through extensive qualitative comparisons with Janus-Pro in **Figure 5 and Figure 10 in the new version of our paper**, we demonstrate the **effectiveness of our method in fine-grained image-text semantic alignment**: compared to the backbone model Janus-Pro, our generated images **consistently and stably align with the requirements specified in the prompt**, and also largely eliminate unnatural distortions (which is a major issue with the backbone model), thereby enhancing semantic consistency and simultaneously improving aesthetic quality compared to the backbone model.
> >
> > Consequently, as aesthetics is not a critical metric for evaluating the effectiveness of the core methodology, we **did not specifically endeavor to enhance the aesthetic quality** to approach the level of high-aesthetic image generation works like SD3 and FLUX in our previously submitted version of Janus-FocusDiff. For instance, we did not perform quality tuning, nor did we introduce any aesthetic-related reward scores during RL, which would substantially enhance the aesthetic effect for image generation.
> >
> > &nbsp;
> >
> > **(3) Improving Aesthetic Quality in the New Version**
> >
> > As stated in (2), since our pipeline and training paradigm have been validated as effective to achieve a significant improvement in fine-grained text-image semantic alignment, **the next step is to further enhance aesthetic quality**. And **enhancing aesthetic quality is comparatively easier to achieve**, which can be efficiently resolved by conducting quality tuning using a only small amount of high-quality SFT data.
> >
> > To this end, we introduce GPT-Image-1.5M and NanoBanana-130K as new data sources to construct **a new version of the FocusDiff-Data**, the images within which are generated by **GPT-4o or nanobanana with very high aesthetic quality**. We then train our model on this new version of FocusDiff-Data following our proposed training paradigm (SFT+Pair-GRPO). Since conducting SFT based on these high-quality images can be viewed as a process of quality tuning, **as shown in the last row of Figure 10 in the new version of our paper**, the images generated by our new Janus-FocusDiff model **substantially resolve the aesthetic issue, significantly improving the aesthetic quality**.
> >
> > &nbsp;
> >
> > Once again, we truly appreciate your time and thoughtful consideration throughout the review process. We warmly welcome any further questions or suggestions. Thank you!

---

### Official Review · Reviewer_uS6o · 2025-10-27

**Soundness:** 3
**Presentation:** 2
**Contribution:** 2
**Rating:** 4
**Confidence:** 4

**Summary:**

This paper addresses the issue of fine-grained semantic control in autoregressive (AR) text-to-image models. To highlight this limitation, the authors introduce PairComp, a new benchmark consisting of paired prompts with subtle semantic differences. To solve the problem, they propose FocusDiff, a training paradigm that includes a new dataset of similar text-image pairs (FocusDiff-Data) and an enhanced reinforcement learning algorithm (Pair-GRPO) designed to learn from these differences. The method is applied to the Janus-Pro model, and experiments show improved performance on PairComp and other standard benchmarks.

**Strengths:**

1.	Clear Narrative: The paper presents a clear and logical narrative, effectively identifying a key limitation in AR models through a benchmark and then proposing a targeted solution comprising a new dataset and a RL training algorithm.
2.	Comprehensive Experiments: The evaluation is thorough, demonstrating the method's effectiveness not only on the proposed PairComp benchmark but also on established general-purpose benchmarks, which validates the model's overall capabilities post-training.
3.	In-depth Analysis: The work includes extensive analytical experiments. The rigorous evaluation design, such as using stricter metrics like the Full-Pass Rate (FPR) and employing multiple validation models, highlights the authors' workload.

**Weaknesses:**

1.	[Major] Insufficient Motivation and Positioning: The motivation for the PairComp benchmark feels insufficiently urgent. The challenge of fine-grained control is a known problem. The paper should acknowledge and differentiate its contributions from prior work like Winoground-T2I [1] and EvoGen [2] rather than explain its unique value beyond existing single-prompt benchmarks.
2.	[Major] Incremental Algorithmic Contribution: The novelty of Pair-GRPO over the standard GRPO appears limited. It seems that the prompt pairs are completely decoupled as RL inputs, and the advantage is calculated jointly. How does this approach differ from treating the prompt pairs separately as inputs? Does this experiment correspond to w/o Group Expansion? The description of the ablation experiment is too brief and unclear, making it difficult to see the specific ablation settings.
3.	[Major] Potential Fidelity-Control Trade-off: While the proposed model significantly improves fine-grained control and compositionality, a potential trade-off in image fidelity is observed in Figure 5. The generated images (e.g., the rabbit and bananas) seem to exhibit an over-smoothing effect, losing texture detail and realism compared to the Janus-Pro-7B baseline, which appears to have better fidelity.
4.	[Minor] (1) There are inconsistencies in the capitalization of key terms throughout the manuscript, such as "PairComp" versus "Paircomp." (2) Acronyms are not always defined upon their first use. For instance, PPO is mentioned in Section 3.2 but its full name (Proximal Policy Optimization) is not provided.


References:
[1] Zhu X, Sun P, Wang C, et al. A contrastive compositional benchmark for text-to-image synthesis: A study with unified text-to-image fidelity metrics. arXiv preprint arXiv:2312.02338, 2023.
[2] Han E X, Jin L, Liu X, et al. Progressive compositionality in text-to-image generative models. arXiv preprint arXiv:2410.16719, 2024.

**Questions:**

1.	[For W1] Could the authors elaborate on the unique necessity and contribution of the PairComp benchmark in light of existing compositional evaluation benchmarks such as Winoground-T2I and EvoGen?
2.	[For W2] Can you clarify the mechanism of joint advantage calculation in Pair-GRPO and how it differs from a baseline of processing the prompt pairs independently? Additionally, please provide more details on the setup of ablation.
3.	[For W3] The results in Figure 5 suggest a potential trade-off where improved semantic control comes at the cost of image fidelity (e.g., over-smoothing). Have the authors analyzed or experimented this phenomenon, and could they comment on this apparent trade-off?
4.	[For W4] We kindly ask the authors to review the manuscript for consistency in terminology and to ensure all acronyms are defined upon first use.

---

> ### Author Response · Authors · 2025-11-23
> **Response to Reviewer uS6o Part 1/4**
>
> Thank you for your valuable comments. We are encouraged that **our work is acknowledged for its clear and logical narrative, alongside its thorough evaluation**. We will address your concerns point by point.
>
> > **Weakness1 and Question1: Insufficient Motivation and Positioning**
>
> **A1:** Thank you for your question. Based on the two works you provided, Winoground-T2I is a compositional evaluation benchmark where each case has paired prompts, while EvoGen is dedicated to constructing contrastive pairs for training. Therefore, it seems that Winoground-T2I and our PairComp benchmark, EvoGen and our FocusDiff, share some similarities. It is essential to clarify the unique necessity and contribution of **our PairComp relative to Winoground-T2I**, and **FocusDiff-Data relative to EvoGen**.
>
> **(1.1) Compare PairComp and Winoground-T2I:**
>
> Although both benchmarks feature test cases of paired prompts that appear similar, they differ in the **design motivation**:
>
> The contrastive pairs in Winoground-T2I are specifically designed for **compositional evaluation**. While there may only be a difference of one or two words between the two prompts, **the main semantic meaning of the prompts is different**, leading to the target images with a drastic variation. Here are two typical examples：
> * "Real police officer with real police car" *v.s.* "Toy police officer with toy police car".
> * "A boy jumping towards the fence and the river" *v.s.* "A boy jumping away from the fence and the river"
>
> As shown, the distinguishing words between the prompts are **modifiers of the core subject's semantics**. The difference between "toy" and "real" determines the overall style of the main scene, and the difference between "towards" and "away" dictates the subject's positional orientation. Consequently, the resulting target images are vastly different, which allows **Winoground-T2I to evaluate whether the model has learned the semantics of these core modifying words.**
>
> ---
>
> In contrast, PairComp not only builds upon the compositional evaluation, but also assesses whether the images generated by the model attend to **fine-grained semantic details that are often overlooked beyond the primary semantics of a prompt**. For instance, consider the following two prompts:
> * "A man in a blue jacket with buttons holds a coffee cup in a park."
> * "A man in a blue jacket with a zipper holds a coffee cup in a park."
>
> The core semantic content of both prompts is largely identical, depicting a man in a blue jacket holding a coffee cup in a park. However, a **subtle, fine-grained difference** exists concerning whether the jacket is fastened with buttons or a zipper. Thus, PairComp requires models to generate the correct subject semantics while also **attending to these subtle details and correctly generating these easily overlooked aspects.**

---

> > ### Author Response · Authors · 2025-11-23
> > **Response to Reviewer uS6o Part 2/4**
> >
> > **(1.2) Compare FocusDiff-Data and EvoGen:**
> >
> > Although both FocusDiff-Data and EvoGen appear dedicated to constructing contrastive pairs, they differ in the following fundamental aspects, ***with the qualitative comparisons shown in Appendix Figure 11 of the new version of our paper***:
> >
> >
> > **(1.2.1) Motivations and Design Purpose**
> >
> > The contrastive pairs utilized in **EvoGen** are primarily designed to **permute the core attributes or relational states of entities** to facilitate **compositional learning**. Typical instances involve swapping the subject and object while preserving the predicate, as demonstrated by examples such as:
> > * "A girl chases a dog" vs. "A dog chases a girl."
> > * "A dog is to the left of a pig" vs. "A pig is to the left of a dog."
> >
> > These pairs effectively compel a model to acquire the semantics of relational terms, such as "chase" and "left", which are pivotal to the **overall meaning of a text prompt**. Although prompts like "A girl chases a dog" and "A dog chases a girl" only differ in word order, their **core semantic roles are reversed**, resulting in corresponding ground-truth images that depict **clearly distinct visual narratives**.
> >
> > ---
> >
> > In contrast, **FocusDiff-Data** not only builds upon the compositional learning of relational terms, but also **emphasizes fine-grained semantic details** that are often overlooked beyond the primary semantics of a prompt. For instance, consider the following two prompts:
> > * "A tennis player in a white outfit holding a racket and a tennis ball, with the **ball in her hand**."
> > * "A tennis player in a white outfit holding a racket and a tennis ball, with the **ball in the air**."
> >
> > The core semantic content of both prompts is largely identical, depicting a tennis player engaged in playing tennis. However, a **subtle, fine-grained difference** exists concerning whether the ball is still in the hand or has been tossed into the air. As the capabilities of generative models continue to advance, it becomes increasingly crucial to capture and control subtle details within the prompts for generating images, especially **the minor details beyond the core semantics**, which is a key motivation for the design of our FocusDiff-data.
> >
> > ---
> >
> > **In summary**, EvoGen aims for the model to learn the **compositional semantics**, such as relational terms, which constitute **the core semantics of the prompt**. While FocusDiff builds upon this by also encouraging the model to attend to **minor, easily overlooked details beyond the core semantics**, thus ensuring the generated image perfectly aligns with the prompt's requirements.
> >
> > **(1.2.2) Construction Pipeline and Data Characteristics**
> >
> > Driven by divergent goals, EvoGen and FocusDiff-Data adopt fundamentally different data-construction pipelines, which in turn shape the distinct characteristics of their respective contrastive pair data.
> >
> > As EvoGen is designed for the compositional learning of core semantics—such as entity attributes or relational states—local consistency (e.g., subject identity, background details) is therefore irrelevant for its training objective. The sole requirement is that the two generated images faithfully render the core semantic contrast encoded in the paired prompts.
> >
> > Consequently, EvoGen follows a **prompt-first paradigm**. The process begins by engineering semantically contrasting prompt pairs and then synthesizing the corresponding images using generative models (e.g., SD3). However, it often results in **poor consistency between the two images in a pair**, such as the background and subject IP attributes.
> >
> > ---
> >
> > In contrast, FocusDiff targets the control of fine-grained semantics, thus necessitating the model to infer how subtle changes in textual tokens lead to specific, minute changes in the visual images. This demands near-identical images whose only deviations are localized and subtle.
> >
> > We therefore implement an **image-first pipeline**. We collect numerous pairs from existing image editing datasets **featuring before-and-after-editing images that only exhibit localized, subtle differences**. We then generate style-consistent captions for these pairs with differing words to highlight the subtle image differences.
> >
> > ---
> >
> > **In summary**, EvoGen employs a prompt-first pipeline, resulting in significant **inconsistencies between the two images within a pair**. In contrast, our FocusDiff-Data utilizes an image-first pipeline, where the **two images within a pair only exhibit localized, subtle differences with strong consistency**.

---

> > > ### Author Response · Authors · 2025-11-23
> > > **Response to Reviewer uS6o Part 3/4**
> > >
> > > > **Weakness2 and Question2: Incremental Algorithmic Contribution**
> > >
> > > Thank you for your question. We will elaborate on the design motivation and performance superiority of PairGRPO compared to GRPO.
> > >
> > > **(2.1) Design Motivation**
> > >
> > > First, we acknowledge that our objective is not to propose a brand-new RL algorithm. Instead, we aim to introduce a **suitable algorithmic refinement that aligns more closely with our core motivation and design principles**.
> > >
> > > And the novelty of our paper lies in our high-level design principles: transition from aligning a single text-image pair for vision-language global alignment to learning the subtle differences between two similar text-image pairs for fine-grained vision-language alignment. This naturally leads to our main contribution: **the innovative data construction pipeline, the respective contributions of which are acknowledged by reviewers TGyE and 8cu7**.
> > >
> > > Given the key contributions in design principles and data construction, algorithmic refinement should align with the core motivation for practical improvement rather than solely pursuing innovation. On this basis, Pair-GRPO is essentially **closely aligned with our high-level design principles**, upgrading vanilla GRPO with two main improvements:
> > >
> > > **(a) Expanding the Group Concept:** as stated in lines 249-263 of our paper, while vanilla GRPO considers $G$ responses from the same prompt as a group, we expand this to include $2\times G$ responses from a pair of similar prompts. As each prompt pair shares highly similar global semantics, **it ensures comparability within the group, making group-relative advantage computation meaningful for a pair of different prompts**. Thus, it is more effective than vanilla GRPO in guiding the model to learn fine-grained semantic differences within a similar prompt pair.
> > >
> > > **(b) Shifting Focus from Exploitation to Exploration:** In Pair-GRPO, we introduce ground-truth images from FocusDiff-Data as the supervision signals with a dynamic adjustment of the ground-truth inclusion probability, as stated in lines 264-294 in our paper. These ground-truth images serve as the exploitation, **guiding the model to better explore reasoning paths to prevent convergence collapse with dual roles**. When paired with corresponding prompts, they serve as positive examples reflecting a positive reward trend; when paired with a similar but mismatched prompt, they serve as negative examples that provide suppressive signals. Coupled with our dynamic adjustment mechanism, these images offer strong supervisory guidance early in training and gradually transition the model toward autonomous exploration.
> > >
> > > **(2.2) Performance Superiority with Ablation Study**
> > >
> > > Here we present a detailed description of our ablation study, the summarized results of which are shown in Table 4 of our paper. Since PairGRPO incorporates two main improvements over GRPO, ablation study is designed across three settings:
> > >
> > > **(a) w/o Group Expanding:** We remove "Expanding the Group Concept", considering only images generated from a single prompt as a group.
> > >
> > > **(b) w/o GT Image:** We remove "Shifting Focus from Exploitation to Exploration", no longer inject ground-truth images from the FocusDiff-Data during RL training.
> > >
> > > **(c) Vanilla GRPO:** We remove both "Expanding the Group Concept" and "Shifting Focus from Exploitation to Exploration", degrading Pair-GRPO to vanilla GRPO.
> > >
> > > The experimental results of ablation study on PairComp and GenEval are given in  ***Table-Rebuttal-1 and Table-Rebuttal-2***. We can see that **Pair-GRPO consistently outperforms other ablated algorithms including vanilla GRPO**. It confirms that **both proposed improvements in Pair-GRPO are highly effective**.
> > >
> > > &nbsp;
> > >
> > > **Table-Rebuttal-1:** Ablation study on PairComp with InternVL2.5.
> > > |Methods|Overall Appear-$S_a$|Overall Appear-$S_g$|Color-$S_a$|Color-$S_g$|Counting-$S_a$|Counting-$S_g$|Position-$S_a$|Position-$S_g$|Style&Tone-$S_a$|Style&Tone-$S_g$|Text-$S_a$|Text-$S_g$|Average-$S_a$|Average-$S_g$|
> > > |-|-|-|-|-|-|-|-|-|-|-|-|-|-|- |
> > > |Janus-Pro-7B| 82.3|75.6|95.7|94.0|52.7|47.1|69.4|63.9|92.0|88.7|60.8|53.2|75.5|70.4 |
> > > |Janus-FocusDiff-7B|**85.4**|**82.4**|**97.8**|**97.7**|**71.0**|**69.0**|**75.9**|**74.0**|**94.3**|**93.9**|**85.3**|**83.8**|**85.0**|**83.5** |
> > > |w/o Group Expanding|84.6|79.8|97.2|96.0|64.0|60.0|72.7|69.5|93.5|92.0|86.0|81.5|83.0|79.8|
> > > |w/o GT Image|84.9|81.3|97.6|97.1|70.0|67.0|75.2|72.8|93.5|92.6|83.4|81.1|84.1|82.0|
> > > |Vanilla GRPO|83.6|77.6|96.5|95.0|60.5|56.0|71.0|68.1|92.0|89.9|80.6|72.6|80.7|76.6|
> > >
> > > **Table-Rebuttal-2:** Ablation study on GenEval.
> > > |Methods|Overall|SingObj|TwoObj|Counting|Color|Pos.|ColorAttr|
> > > |-|-|-|-|-|-|-|-|
> > > |Janus-Pro-7B|0.80|0.99|0.89|0.59|0.90|0.79|0.66|
> > > |Janus-FocusDiff-7B|**0.87**|**0.99**|**0.96**|**0.67**|**0.94**|**0.87**|**0.77** |
> > > |w/o Group Expanding|0.84|0.99|0.94|0.62|0.92|0.84|0.73|
> > > |w/o GT Image|0.85|0.99|0.95|0.64|0.93|0.85|0.74|
> > > |Vanilla GRPO|0.83|0.99|0.92|0.61|0.91|0.83|0.72|

---

> > > > ### Author Response · Authors · 2025-11-23
> > > > **Response to Reviewer uS6o Part 4/4**
> > > >
> > > > > **Weakness3 and Question3: Potential Fidelity-Control Trade-off**
> > > >
> > > > **A3:** We appreciate you pointing out this critical issue regarding the aesthetic quality of the generated images. First, we include a qualitative comparison in ***Appendix Figure 10 in the new version of our paper***. It showcases multiple generated images from the same prompt using both Janus-Pro and Janus-Pro-FocusDiff to illustrate their respective aesthetic effects. A key finding is that **images generated by Janus-Pro frequently exhibit noticeable artifacts and unnatural structures**, such as deformed cows. While these defects are significantly improved in those generated by Janus-Pro-FocusDiff, this indicates that **the images generated by our model generally achieve better aesthetic quality**.
> > > >
> > > >
> > > > However, as you observed, many cases from our model show an over-smoothing effect, resulting in a tendency to generate images with a painting-like style. A significant contributing factor to this is that **our training data source from earlier image editing datasets such as Anyedit and Ultraedit** (we also conduct fine-tuning on these data before RL). Due to the suboptimal editing pipelines used to construct these datasets, the images within them often **suffer from artifacts like a loss of texture detail and realism**. Therefore, performing SFT based on such data subsequently leads to the over-smoothing issues in our generated images.
> > > >
> > > > Moreover, over the past few months, some high-aesthetic-quality editing datasets, such as GPT-Image-1.5M and nanobanana-130K, have emerged. We have continued to iterate and refine our FocusDiff-Data using these superior datasets. Based on the new version of FocusDiff-Data, we further perform  SFT and RL training on Janus-Pro. As a result, **the new version of Janus-FocusDiff substantially resolves the over-smoothing issue**, as shown in ***Appendix Figure 10 in the new version of our paper***.
> > > >
> > > > &nbsp;
> > > >
> > > > > **Weakness4 and Question4: Review the manuscript for consistency in terminology and to ensure all acronyms are defined upon first use.**
> > > >
> > > > **A4:** Thank you for pointing out the inconsistencies in terminology and the oversight regarding the definition of acronyms upon their first use. We sincerely apologize for these errors. We have fully reviewed the manuscript and incorporated the corresponding revisions into the newly submitted PDF.

---

### Official Review · Reviewer_8cu7 · 2025-10-30

**Soundness:** 2
**Presentation:** 1
**Contribution:** 2
**Rating:** 4
**Confidence:** 4

**Summary:**

This paper tackles the problem of improving compositionality in autoregressive text-to-image generative models. The authors first introduce a contrastive dataset, where each example consists of two text–image pairs that differ subtly in a specific attribute or semantic aspect. Building on this idea, they also construct a benchmark designed to evaluate models’ sensitivity to fine-grained compositional differences between prompts. To enhance AR models, the paper further extends the GRPO optimization framework by proposing Pair-GRPO, which generalizes GRPO to operate on grouped pairs of prompts, leveraging the proposed contrastive dataset to improve compositional understanding during training.

**Strengths:**

- The paper tackles a highly important problem, improving compositionality in AR text-to-image generative models, which is crucial for achieving more reliable and interpretable generation.
- A key contribution is the contrastive pair dataset construction: rather than synthetically generating contrastive pairs (as done in prior work, which often introduces inconsistencies), the authors leverage existing high-quality image editing datasets and apply additional filtering strategies to ensure semantic precision and data quality.
- The paper presents experimental results, including both quantitative and qualitative evaluations that demonstrate the effectiveness of the proposed approach.
- For the quantitative evaluation, they have covered multiple benchmarks that assess the compositional performance of T2I models.

**Weaknesses:**

- The writing quality requires significant improvement. The paper often lacks clarity and coherence in some parts. For instance, in the introduction, the arithmetic and geometric evaluation metrics are introduced abruptly without proper explanation, which can confuse readers. In addition, several notation errors and inconsistencies are present (e.g., around lines 136 and 243), which further detract from readability.
- The evaluation methodology is overly simplistic. The authors rely on a single VLM-based binary question (“Does this image match the prompt?”), which provides a limited assessment of compositional alignment. Prior works have adopted more fine-grained and structured evaluations, such as decomposing prompts into sub-questions to test specific compositional aspects, leading to more reliable measurements.
- The contrastive pair construction approach, while effective, is not particularly novel. Similar ideas have been explored in earlier works such as [1], which also utilized contrastive data to enhance compositional learning.
- The paper claims state-of-the-art performance on benchmarks like GenEval, but important baselines such as Flow-GRPO [2] are omitted, despite reporting significantly stronger results.
- For the T2I-CompBench, it would be beneficial to include all categories (e.g., spatial, numeracy, ...) to provide a more comprehensive comparison.

[1] Han, Evans Xu, et al. "Progressive compositionality in text-to-image generative models." arXiv preprint arXiv:2410.16719 (2024).

[2] Liu, Jie, et al. "Flow-grpo: Training flow matching models via online rl." arXiv preprint arXiv:2505.05470 (2025).

**Questions:**

-

---

> ### Author Response · Authors · 2025-11-23
> **Response to Reviewer 8cu7 Part 1/3**
>
> Thank you for the valuable comments. We are encouraged that **the contrastive pair dataset construction is acknowledged as a key contribution**. We will address your concerns point-by-point.
>
> > **Weakness1: The writing quality requires improvement.**
>
> **A1:** We sincerely apologize for any shortcomings in the initial presentation. According to your suggestions, **we have made several revisions to enhance the clarity and readability of the manuscript**. For instance, in the introduction, we now provide relevant explanations for both arithmetic and geometric evaluations before referencing their application. Furthermore, near lines 136 and 243, we add clarifying notes on the notational symbols and address several inconsistencies to improve the overall readability.
>
> &nbsp;
>
> > **Weakness2: The evaluation methodology is overly simplistic.**
>
> **A2:** Thank you for the valuable feedback. We will provide explanations from two perspectives: **PairComp benchmark evaluation** and **RL reward calculation**.
>
> As for **benchmark evaluation**, prior benchmarks such as DPG-Bench decompose prompts into sub-questions for evaluation because their prompts consist of long captions, often exceeding 50 words. In contrast, our **PairComp benchmark utilizes shorter prompts**, typically under 20 words. For prompts of this length, we find VLMs such as **InternVL2.5-26B and QwenVL2.5-72B demonstrate sufficient capability to precisely judge fine-grained text-image alignment with a single binary question**, without the need for prompt decomposition.
>
> To validate this, we follow DPG-bench to decompose prompts into sub-questions for PairComp evaluation, with the results presented in ***Table-Rebuttal-2 below and Table 8 in the new version of paper*** (***Table-Rebuttal-3*** shows the results with the original evaluation strategy). **The relative performance ranking of most models remains largely unchanged with the new strategy**, with our model consistently outperforming other baselines.
>
> Moreover, we randomly sample 50 cases to compare the consistency (Pearson correlation coefficient) between both evaluation strategies and human rating. As shown in ***Table-Rebuttal-1***, we find **prompt decomposition offers virtually no improvement in evaluation reliability for PairComp** compared to using a single binary question. Thus, we adopt the simplified approach of a single VLM-based binary question in our experiments.
>
> Furthermore, during RL, we also leverage VLM to evaluate text-image alignment for **reward calculation**. Specifically, as outlined ***in Lines 824-828 of our paper, for long prompts in FocusDiff-Data, we decompose the prompt into sub-questions which are then presented to the VLM, to get a more accurate reward score***; for short captions, we directly instruct the VLM with a single binary question to compute rewards.
>
> &nbsp;
>
> **Table-Rebuttal-1:** Pearson correlation coefficient with human evaluations for each evaluation strategy based on InternVL2.5.
> |Evaluation Strategy|Pearson-r with human rating↑|
> |-|-|
> |Single Binary Question|0.77|
> |Prompt Decomposition|0.78|
>
> **Table-Rebuttal-2:** Evaluation results on PairComp based on InternVL2.5, when decomposing the prompts into sub-questions.
> |Models|Overall Appear-$S_a$| Overall Appear.-$S_g$|Color-$S_a$|Color-$S_g$|Counting-$S_a$|Counting-$S_g$|Position-$S_a$|Position-$S_g$|Style&Tone-$S_a$|Style&Tone-$S_g$|Text-$S_a$|Text-$S_g$|Average-$S_a$|Average-$S_g$|
> |-|-|-|-|-|-|-|-|-|-|-|-|-|-|-|
> |LLamaGen|60.8|56.3|69.4|64.4|48.1|42.1|50.4|43.7|75.5|66.8|27.4|20.9|55.3|49.0|
> |Show-o|75.9|72.6|89.1|87.0|62.2|58.9|52.0|47.4|92.4|89.3|42.1|34.2|68.9|64.9|
> |Bagel-Think|82.3|79.8|92.1|90.6|72.7|69.8|72.7|67.5|91.1|87.4|65.1|58.4|79.3|75.6|
> |Janus-Pro-7B|84.1|81.3|96.4|94.8|55.4|49.2|72.6|68.2|94.5|91.2|65.4|58.5|78.1|73.9|
> |SD3|87.0|85.2|96.1|95.4|**78.6**|**75.8**|74.8|71.6|90.5|87.3|**95.0**|**94.1**|87.0|84.9|
> |Janus-FocusDiff-7B|**89.3**|**87.9**|**98.6**|**98.3**|72.0|70.1|**78.1**|**75.2**|**95.4**|**94.9**|88.1|86.8|**86.9**|**85.5**|
>
> **Table-Rebuttal-3:** Evaluation results on PairComp based on InternVL2.5-26B, when directly leveraging a single binary question.
> | Models|Overall Appear-$S_a$| Overall Appear-$S_g$|Color-$S_a$|Color-$S_g$|Counting-$S_a$|Counting-$S_g$| Position-$S_a$|Position-$S_g$|Style&Tone-$S_a$|Style&Tone-$S_g$|Text-$S_a$|Text-$S_g$|Average-$S_a$|Average-$S_g$|
> |-|-|-|-|-|-|-|-|-|-|-|-|-|-|-|
> |LLamaGen|53.5|45.4|67.0|61.2|45.3|39.5|42.1|35.4|68.8|60.1|18.0|12.0|49.1|42.3|
> |Show-o|68.5|62.2|87.2|85.0|58.2|55.2|45.2|40.6|87.8|84.7|34.9|26.8|63.6|59.1|
> |Bagel-Think|81.8|78.1|90.2|88.7|69.2|66.6|68.8|62.6|87.9|84.2|60.2|53.1|76.4|72.2|
> |Janus-Pro-7B|82.3|75.6|95.7|94.0|52.7|47.1|69.4|63.9|92.0|88.7|60.8|53.2|75.5|70.4|
> |SD3|82.5|77.0|95.4|94.6|**74.0**| **70.3**|71.9|68.5|89.4|86.2|**93.1**|**92.0**|84.4|81.4|
> |Janus-FocusDiff-7B|**85.4**|**82.4**|**97.8**|**97.7**|71.0|69.0|**75.9**|**74.0**|**94.3**|**93.9**|85.3|83.8|**85.0**|**83.5**|

---

> > ### Author Response · Authors · 2025-11-23
> > **Response to Reviewer 8cu7 Part 2/3**
> >
> > > **Weakness3: The contrastive pair construction approach, while effective, is not particularly novel**
> >
> > **A3:** Thank you for your insightful comment. As you've noted, EvoGen, introduced in "Progressive Compositionality in Text-to-Image Generative Models", also constructs contrastive pairs. Therefore, it is essential to clarify the unique necessity and contribution of our FocusDiff-Data relative to EvoGen. Although both approaches appear dedicated to constructing contrastive pairs, they differ in the following fundamental aspects, ***with the qualitative comparisons shown in Appendix Figure 11 of the new version of our paper***:
> >
> > **(3.1) Motivations and Design Purpose**
> >
> > The contrastive pairs utilized in **EvoGen** are primarily designed to **permute the core attributes or relational states of entities** to facilitate **compositional learning**. Typical instances involve swapping the subject and object while preserving the predicate, as demonstrated by examples such as:
> > * "A girl chases a dog" vs. "A dog chases a girl."
> > * "A dog is to the left of a pig" vs. "A pig is to the left of a dog."
> >
> > These pairs effectively compel a model to acquire the semantics of relational terms, such as "chase" and "left", which are pivotal to the **overall meaning of a text prompt**. Although prompts like "A girl chases a dog" and "A dog chases a girl" only differ in word order, their **core semantic roles are reversed**, resulting in corresponding ground-truth images that depict **clearly distinct visual narratives**.
> >
> > ---
> >
> > In contrast, **FocusDiff-Data** not only builds upon the compositional learning of relational terms, but also **emphasizes fine-grained semantic details** that are often overlooked beyond the primary semantics of a prompt. For instance, consider the following two prompts:
> > * "A tennis player in a white outfit holding a racket and a tennis ball, with the **ball in her hand**."
> > * "A tennis player in a white outfit holding a racket and a tennis ball, with the **ball in the air**."
> >
> > The core semantic content of both prompts is largely identical, depicting a tennis player engaged in playing tennis. However, a **subtle, fine-grained difference** exists concerning whether the ball is still in the hand or has been tossed into the air. As the capabilities of generative models continue to advance, it becomes increasingly crucial to capture and control subtle details within the prompts for generating images, especially **the minor details beyond the core semantics**, which is a key motivation for the design of our FocusDiff-data.
> >
> > ---
> >
> > **In summary**, EvoGen aims for the model to learn the **compositional semantics**, such as relational terms, which constitute **the core semantics of the prompt**. While FocusDiff builds upon this by also encouraging the model to attend to **minor, easily overlooked details beyond the core semantics**, thus ensuring the generated image perfectly aligns with the prompt's requirements.
> >
> > &nbsp;
> >
> > **(3.2) Construction Pipeline and Data Characteristics**
> >
> > Driven by divergent goals, EvoGen and FocusDiff-Data adopt fundamentally different data-construction pipelines, which in turn shape the distinct characteristics of their respective contrastive pair data.
> >
> > As EvoGen is designed for the compositional learning of core semantics—such as entity attributes or relational states—local consistency (e.g., subject identity, background details) is therefore irrelevant for its training objective. The sole requirement is that the two generated images faithfully render the core semantic contrast encoded in the paired prompts.
> >
> > Consequently, EvoGen follows a **prompt-first paradigm**. The process begins by engineering semantically contrasting prompt pairs and then synthesizing the corresponding images using generative models (e.g., SD3). However, it often results in **poor consistency between the two images in a pair**, such as the background and subject IP attributes.
> >
> > ---
> >
> > In contrast, FocusDiff targets the control of fine-grained semantics, thus necessitating the model to infer how subtle changes in textual tokens lead to specific, minute changes in the visual images. This demands near-identical images whose only deviations are localized and subtle.
> >
> > We therefore implement an **image-first pipeline**. We collect numerous pairs from existing image editing datasets **featuring before-and-after-editing images that only exhibit localized, subtle differences**. We then generate style-consistent captions for these pairs with differing words to highlight the subtle image differences.
> >
> > ---
> >
> > **In summary**, EvoGen employs a prompt-first pipeline, resulting in significant **inconsistencies between the two images within a pair**. In contrast, our FocusDiff-Data utilizes an image-first pipeline, where the **two images within a pair only exhibit localized, subtle differences with strong consistency**.

---

> > > ### Author Response · Authors · 2025-11-23
> > > **Response to Reviewer 8cu7 Part 3/3**
> > >
> > > > **Weakness4: Important baselines such as Flow-GRPO are omitted.**
> > >
> > > **A4:** We acknowledge that Flow-GRPO achieved superior performance to our model in GenEval, and apologize for our previous imprecise statements and have revised them in the paper. A more accurate assertion is that **our method significantly enhances the text-to-image performance of the Janus-Pro backbone model, demonstrating markedly better results on benchmarks such as GenEval and PairComp, and even surpassing many inherently stronger image generation models.**
> > >
> > > Furthermore, FlowGRPO possesses a different architecture with Janus-Pro and its open-sourced checkpoint is **specifically optimized for the GenEval task**, whereas **our method uses general FocusDiff-Data for RL without task-specific optimization**. For instance, when we evaluate the performance of Flow-GRPO on the PairComp benchmark, as presented in ***Table-Rebuttal-4 below***, **our model exhibits stronger performance than Flow-GRPO**.
> > >
> > > **Table-Rebuttal-4:** Performance Comparison between Flow-GRPO and Janus-FocusDiff on PairComp with InternVL2.5-26B as the evaluator.
> > > | Models|Overall Appear.-$S_a$|Overall Appear.-$S_g$|Color-$S_a$|Color-$S_g$|Counting-$S_a$|Counting-$S_g$|Position-$S_a$|Position-$S_g$|Style&Tone-$S_a$|Style&Tone-$S_g$|Text-$S_a$|Text-$S_g$|Average-$S_a$|Average-$S_g$ |
> > > |-|-|-|-|-|-|-|-|-|-|-|-|-|-|-|
> > > |Flow-GRPO| 82.3|77.8|95.3|95.2|61.5|57.2|70.6|67.0|90.1|88.1|**93.2**|**92.1**|82.2|79.6 |
> > > |Janus-FocusDiff| **85.4**|**82.4**|**97.8**|**97.7**|**71.0**|**69.0**|**75.9**|**74.0**|**94.3**|**93.9**|85.3|83.8|**85.0**|**83.5** |
> > >
> > > &nbsp;
> > >
> > > > **Weakness5: For the T2I-CompBench, it would be beneficial to include all categories.**
> > >
> > > **A5:** Thank you for your suggestions! We initially reported results only for the three subtasks (Color, Shape, and Texture) because many preceding baseline papers focused solely on these subtasks. **The paper of T2I-CompBench states that it contains a total of six subtasks, including color, shape, texture, spatial, non-spatial and complex**.
> > >
> > > We now additionally report the evaluation results on another three subtasks, i.e., spatial, non-spatial and complex. As shown in the following ***Table-Rebuttal-5 and Table 9 in the new version of our paper***, compared to the backbone model Janus-Pro-7B, our **model achieves significant performance improvements and also outperforms other baseline methods**.
> > >
> > > **Table-Rebuttal-5:** Performance Comparison on all six subtasks of T2I-CompBench.
> > > |Models| Color| Shape|Texture| Spatial| Non-Spatial| Complex|
> > > |-|-|-|-|-|-|-|
> > > |FLUX.1-dev|0.74|0.57|0.69|0.29|0.31|0.37|
> > > |Show-o|0.56|0.41|0.46|0.20|0.30|0.29|
> > > |Show-o+PARM|0.75|0.56|0.66|0.29|0.31|0.37|
> > > |Janus-Pro-7B|0.64|0.35|0.49|0.21|0.31|0.36|
> > > |T2I-R1|0.81|0.59|0.72|0.34|0.31|0.40|
> > > |Janus-FocusDiff-7B|**0.83**|**0.61**|**0.74**|**0.35**|**0.33**|**0.42**|

---

### Official Review · Reviewer_TGyE · 2025-11-03

**Soundness:** 3
**Presentation:** 3
**Contribution:** 3
**Rating:** 6
**Confidence:** 4

**Summary:**

Existing autoregressive text-to-image models fail at fine-grained semantic control (e.g., color, position, counting). This paper introduces a new framework to address this weakness, namely:
1. PairComp, a benchmark built on minimal pairs of prompts.
2. A new metric, the geometric mean $s_g$, to penalize models that fail on one half of a pair.
3. FocusDiff-Data, a 200k-sample dataset of contrastive text-image tuples $(I_1, T_1, I_2, T_2)$)created by applying a VLM to image editing datasets.
4. Pair-GRPO, an RL algorithm adapted from GRPO (Shao et al., 2024) which expands the advantage-calculation group to include samples from both contrastive prompts $T_1$ and $T_2$ and uses a ground-truth-guided curriculum for stable training.
5. Janus-FocusDiff, which shows gains over its backbone (Janus-Pro) on PairComp and other T2I benchmarks.

**Strengths:**

1. The PairComp benchmark is a strong contribution. The design is a more direct and rigorous probe than existing single-prompt compositional benchmarks. The justification for the geometric mean ($s_g$) (l. 137) is sound and identifies an important failure mode (unstable generation).
2. The pipeline for creating FocusDiff-Data (Sec 3.1, Appendix C) is innovative. Using image editing datasets as a source for contrastive pairs is a clever solution to the data-sourcing problem, and the resulting dataset is a valuable contribution.
3. The adaptation of GRPO, an algorithm from the language reasoning domain, to the problem of autoregressive visual token generation is a creative and non-obvious transfer.
4. The core ideas of Pair-GRPO are well-motivated. (1) Expanding the group to include contrastive samples (l. 241) is an intuitive way to force the policy to learn differentiation. (2) Likewise using the ground-truth pairs (including negative examples, e.g., $(T_1, \hat{I}_2)$) as part of a training curriculum (l. 263) is a strong training signal.

**Weaknesses:**

1. Conflating reward model + evaluator. The QA-based reward model for RL is InternVL2.5-26B (l. 820). The primary evaluation model for the PairComp benchmark is also InternVL2.5-26B (l. 124, 216). This is a confound. The RL policy is being optimized to maximize the score of the same model that is used to judge its performance. The gains reported in the main results (Table 1) could represent reward hacking or overfitting to the specific biases of the InternVL2.5-26B evaluator, rather than a true generalizable improvement in alignment.
The authors provide results with a held-out evaluator (Qwen2.5-VL-72B) in Appendix Table 6 (l. 919). These should be in the main paper as the primary evidence for the method’s success.
2. Missing SFT-Only Baseline. The pipeline consists of two stages: (1) SFT on FocusDiff-Data (l. 194, 812), followed by (2) RL with Pair-GRPO (l. 195). The main comparison (e.g., in Table 1) is between the original Janus-Pro-7B (Row 1) and the final Janus-FocusDiff-7B (Row 2), which has had both SFT and RL. Where is the baseline with Janus-Pro-7B + SFT on FocusDiff-Data (but no RL stage)? Without this baseline, it is impossible to disentangle the gains from the SFT stage vs. the novel Pair-GRPO stage. The SFT on this highly-curated contrastive dataset might be responsible for most of the improvement, with the complex RL algorithm providing only marginal benefit. The paper's central claim is about the effectiveness of Pair-GRPO, but the experiments do not provide the necessary evidence to support this claim over the simpler alternative (SFT is sufficient).
3. Justification for Pointwise vs. Pairwise Reward. The paper adopts a pointwise reward model (l. 821-824), where each (image, text) pair receives an absolute score $S(I, T)$. The GRPO objective (Eq. 1) then normalizes these absolute scores.
To my understanding, concurrent research (e.g., Pref-GRPO [2508.20751]) has identified that this pointwise score-maximization paradigm in T2I-RL is unstable and prone to "illusory advantages" (where tiny, meaningless score differences are amplified) and reward hacking.
These works propose pairwise preference rewards (i.e., "is $I_a$ better than $I_b$ for $T$?") as a more robust and stable signal that better aligns with human judgment. The paper cites other RL-for-T2I methods (T2I-R1, l. 876) but misses this work. It should justify why pointwise-reward GRPO is sufficient and not subject to the same instabilities that this other work identifies.
4. Nit. Naming the paper “focusdiff” is confusing. The method is applied to an autoregressive model (Janus-Pro) and uses Reinforcement Learning (RL). The "Diff" suffix is associated with Diffusion models in the current ML literature, which this work is not. This will cause confusion.

**Questions:**

1. Why were the main results (Table 1) reported using the same model for reward and evaluation?
2. Can you provide the missing baseline: Janus-Pro-7B fine-tuned only on the FocusDiff-Data SFT set? So we can properly attribute the performance gains to the SFT stage vs. the Pair-GRPO stage.
3. Can you justify your choice of a pointwise reward model for GRPO over a pairwise preference-based model (given the known instability issues of the former that have been addressed by concurrent work)?

---

> ### Author Response · Authors · 2025-11-23
> **Response to Reviewer TGyE Part 1/2**
>
> Thank you for your valuable suggestions and constructive advice! We are encouraged to see that the **PairComp benchmark is a strong contribution**, **the pipeline for creating FocusDiff-Data is innovative**, and **the core ideas of Pair-GRPO are well-motivated**. Next, we will address your concerns point-by-point.
>
> > **Weakness1 and Question1: Conflating reward model + evaluator.**
>
> **A1:** Thank you for pointing out this issue! In the PairComp evaluation, we leverage both Qwen2.5-VL-72B and InternVL2.5-26B as the VLM evaluator. Strictly speaking, conflating the reward model and the evaluator may lead to reward hacking rather than a true generalizable improvement in alignment. Therefore, the Qwen2.5-VL evaluation results should be considered as the main results. Therefore, in the new version of the paper, we have also ***relocated Table 6 from the Appendix to the main body of the paper as Table 2, presenting the Qwen2.5-VL-72B evaluation as the main result.***
>
>
> Furthermore, it is worth noting that, comparing the results between the Qwen2.5-VL and InternVL2.5 evaluation, **the relative performance ranking of most models remains largely unchanged**. And **on both evaluators**, **Janus-FocusDiff continues to outperform all baselines except Qwen-Image**, with the average performance on PairComp outperforming some leading models such as HunyuanImage-2.1, OmniGen2 and Bagel. It demonstrates the **true effectiveness** of our method rather than performance gains achieved through reward hacking.
>
> &nbsp;
>
> > **Weakness2 and Question2: Missing SFT-Only Baseline.**
>
> **A2:** We apologize for omitting the SFT-only baseline results in the previous manuscript. ***These results are now provided in Table 1, Table 2, and Table 3 of the new version of our paper*** (also presented in ***Table-Rebuttal-1*** below for immediate reference).
>
> We can see that **performance improvement from the SFT phase is very incremental**, with the majority of the performance gains **being attributable to the subsequent reinforcement learning** stage. We then clarify the necessity and limitations of the SFT phase:
>
> **(1)** First, we find that the images generated by Janus-Pro frequently exhibit noticeable artifacts and unnatural structures, as shown in ***Appendix Figure 10 in the new version of our paper***. In contrast, the images curated within our FocusDiff-Data dataset present an overall acceptable aesthetic quality thanks to our visual quality filtering pipeline. Consequently, **following SFT on FocusDiff-Data, we significantly boost the aesthetic quality of the generated images**, which is challenging to achieve solely through RL on Janus-Pro.
>
> **(2)** Despite the benefits of SFT in enhancing aesthetic quality, as for semantic alignment, we find that "SFT Memorizes, RL Generalizes". Given that the backbone, Janus-Pro, has already undergone extensive image-text alignment during pre-training, **it is challenging to achieve further significant performance improvement through SFT alone**. Conversely, utilizing our improved GRPO algorithm and FocusDiff-Data, RL training successfully recasts the autoregressive modeling of visual tokens into a process of genuine reasoning. It effectively boosts the model's generalization capabilities for image generation, leading to a significant performance improvement on benchmarks such as PairComp and Geneval.
>
> &nbsp;
>
> **Table-Rebuttal-1:** Performance of SFT-only baseline in GenEval, DPG-Bench and PairComp
> |Methods| GenEval Overall|DPG-Bench Avg|PairComp (InternVL2.5) Average-$S_a$ |PairComp (InternVL2.5) Average-$S_g$ |PairComp (Qwen2.5-VL) Average-$S_a$ |PairComp (Qwen2.5-VL) Average-$S_g$|
> |--|--|--|--|--|--|--|
> | Janus-Pro-7B|0.80 |84.17|75.5|70.4|63.6|54.1|
> |Janus-Pro-7B+SFT| 0.81 |84.28|77.6|73.0|66.8|58.1|
> | Janus-FocusDiff|**0.87**|**85.31**|**85.0**|**83.5**|**74.8**|**69.9**|

---

> > ### Author Response · Authors · 2025-11-23
> > **Response to Reviewer TGyE Part 2/2**
> >
> > > **Weakness3 and Question3: Justification for Pointwise vs. Pairwise Reward.**
> >
> > **A3:** We apologize for the oversight in not comparing our method against Pref-GRPO and pairwise rewards. This omission was likely due to its publication on arXiv only a few weeks prior to the ICLR submission deadline, which restricted the time available for comprehensive integration and comparison. We appreciate your valuable suggestions and we have conducted a **three-pronged experimental analysis**.
> >
> > **(3.1) Training Stability Comparison between vanilla GRPO and Pair-GRPO**
> >
> > GRPO is well-known for its inherent training instability. However, as detailed in lines 480–485 of our paper, compared to vanilla GRPO, our proposed Pair-GRPO **not only achieves superior performance but also significantly enhances the stability and convergence of the RL training**. Specifically, ***Figure 7(c) in our paper*** illustrates the reward trends across training steps for both vanilla GRPO and Pair-GRPO. While vanilla GRPO exhibits fluctuating rewards, **Pair-GRPO demonstrates a steady and consistent reward improvement**, suggesting that Pair-GRPO substantially mitigates the training instability issues associated with pointwise rewards.
> >
> > Of course, we note that Pair-GRPO sometimes still exhibits training instability, and **its overall training stability remains sensitive to the hyperparameter setup**, particularly the learning rate and the weight of the KL-divergence loss. Consequently, as described in lines 841–854 of our paper, **we carefully design the hyperparameters**, which ensures a stable training run without compromising performance.
> >
> > **(3.2) Performance Comparison between Pointwise Reward and Pairwise Reward**
> >
> > Following your suggestion, we conduct experiments with several hyperparameter configurations by **replacing the original pointwise reward with pairwise reward** under the same setup (the same backbone and training data).
> >
> > We find that leveraging the pairwise reward improves training stability compared to the original pointwise reward setup. However, as shown in the ***Table-Rebuttal-2 below and Table 7 in the new version of our paper***, the performance achieved with the optimal hyperparameters under the pairwise reward on PairComp is inferior to the results obtained using the pointwise reward. This suggests that **while pairwise Reward enhances training stability in our task scenario, it currently exhibits a performance deficit compared to Pointwise Reward**, which needs further investigation.
> >
> > **Table-Rebuttal-2:** Performance comparison between pointwise reward and pairwise reward.
> > |Methods| GenEval Overall|DPG-Bench Avg|PairComp (InternVL2.5) Average-$S_a$ |PairComp (InternVL2.5) Average-$S_g$ |PairComp (Qwen2.5-VL) Average-$S_a$ |PairComp (Qwen2.5-VL) Average-$S_g$|
> > |-|-|-|-|-|-|-|
> > |Janus-Pro-7B|0.80|84.17|75.5|70.4|63.6|54.1|
> > |Janus-FocusDiff(Pairwise Reward)| 0.84 |84.90|82.0|79.8|71.3|65.6|
> > |Janus-FocusDiff(Pointwise Reward)|**0.87**|**85.31**|**85.0**|**83.5**|**74.8**|**69.9**|
> >
> > **(3.3) Performance of Pref-GRPO on PairComp**
> >
> > Finally, to further validate our findings regarding the efficacy of pointwise reward, we evaluate the performance of FLUX.1-dev-PrefGRPO, a successful implementation of pairwise reward, on PairComp. As presented in **Table-Rebuttal-3** below, **the average performance of our Janus-FocusDiff consistently surpasses that of FLUX.1-dev-PrefGRPO**, which further supports the conclusion in the previous point.
> >
> > **Table-Rebuttal-3:** Performance comparison between FLUX.1-dev-PrefGRPO and Janus-FocusDiff-7B on PairComp, with Qwen2.5-VL-72B as the evaluator
> > |Methods|Overall Appear.-$S_a$|Overall Appear.-$S_g$|Color-$S_a$|Color-$S_g$|Counting-$S_a$|Counting-$S_g$|Position-$S_a$|Position-$S_g$|Style&Tone-$S_a$|Style&Tone-$S_g$|Text-$S_a$|Text-$S_g$|Average-$S_a$|Average-$S_g$|
> > |-|-|-|-|-|-|-|-|-|-|-|-|-|-|-|
> > |FLUX.1-dev-PrefGRPO|73.0|65.1|95.1|92.2|46.0|38.8|52.4|45.6|84.2|77.8|87.6|83.7 |73.1|67.2|
> > |Janus-FocusDiff-7B|73.8|67.5|96.2|95.0|46.2|39.7|60.6|53.4|94.1|91.1|77.8 |71.4 |74.8|69.9|
> >
> > In conclusion, we still maintain that pairwise reward represents a very promising direction for improvement over pointwise reward. Given the limited time available during the rebuttal, we think more exploration is required to fully unleash the potential of pairwise reward. We once again thank you for the valuable suggestions.
> >
> > &nbsp;
> >
> > > **Weakness4: Naming the paper "FocusDiff" is confusing.**
> >
> > **A4:** We apologize for any confusion caused by the name "FocusDiff", which was originally intended as an abbreviation for "focus on difference", describing that we aim to enhance the fine-grained text-image semantic alignment by focusing on subtle differences between similar text-image pairs.
> >
> > We regret this oversight since "diff" is often interpreted as an abbreviation for "diffusion", and we will try to set a better model name in the future to avoid misunderstanding. Thank you for pointing out this issue!

---

### Author Response · Authors · 2025-11-23
**General Response to All Reviewers**

Dear Reviewers, ACs, and SACs:

We sincerely thank you for the precious time and insightful feedback, which have significantly strengthened our manuscript! Overall, we are encouraged that you find that:

- The paper tackles a **highly important problem**. (*Reviewer 8cu7*)

- The **PairComp benchmark** is a **strong contribution**. The justification for the geometric mean is sound and identifies an important failure mode. (*Reviewer TGyE*)

- The **pipeline for creating FocusDiff-Data** is **innovative** and a **key contribution**. (*Reviewer TGyE, 8cu7*)

- **The core ideas of Pair-GRPO are well-motivated**, representing a creative and non-obvious transfer from the language reasoning domain to autoregressive visual generation. (*Reviewer TGyE*)

- The **comprehensive and thorough experiments**, which include **extensive analytical studies**, effectively demonstrate **the validity of the proposed method**. (*Reviewers 8cu7, uS6o, YYr2*)

&nbsp;

To address the concerns raised by the reviewers, overall, we have conducted several additional experiments:

- We evaluate our SFT-only model on each benchmark to further demonstrate the significant role played by Pair-GRPO.

- We adjust the original pointwise reward to a pairwise reward, finding that while it improves training stability, the performance is not as good as before.

- We changed the evaluation strategy from the original single VLM-based binary question to decomposing prompts into sub-questions, finding that the new evaluation results still support our original conclusion.

- We show more qualitative comparisons between Janus-FocusDiff and Janus-Pro, proving that the aesthetic quality of our model's generated images is superior to the backbone model. At the same time, we continuously optimize our FocusDiff-Data using higher-quality editing datasets, and the latest version of the model achieves better aesthetic quality.

- We add more baselines on PairComp and evaluate the results of all six subtasks in T2I-CompBench.

We have also clarified the following key points:

- We detail the motivation of our work, specifically by providing a detailed comparison with Winoground-T2I and EvoGen in terms of their underlying purpose, data construction pipeline, and data characteristics.

- We further explain the design motivation of Pair-GRPO and its superiority.

- We move the evaluation results with Qwen2.5-VL-72B on PairComp from the original Appendix to the main body of the paper, which are also presented as the main results of PairComp.

- We check and correct some typos and inconsistent symbols throughout the manuscript to enhance readability.

Most experiments and clarifications have already been integrated into the main body or the appendix of our paper. Once again, we sincerely thank all reviewers for the valuable suggestions! We warmly welcome any further questions or suggestions. Discussions are always open! Thank you!

---

*Best regards,*

*ICLR 2026 Conference Submission5219 Authors*

---

### Meta-Review · Area_Chair_CxdA · 2026-01-07

**Summary:**

The paper proposes a recipe to improve fine-grained control of auto-regressive text-to-image generation. The main contribution is the data, called PairComp, a collection of 2 similar images and their captions (generated by InternVL2.5-26B). These similar images are mined from existing image editing datasets. The paper uses this data to perform SFT and RL (Pair GRPO) and shows that this leads to improvement on the Janus-Pro backbone (Pan et al., 2025)

The paper receives 2 ratings of 6 (TGyE, YYr2), and 2 ratings of 4 (8cu7, uS6o). Summary of concerns:
1. Significance of data and the data generation pipeline due to unclear positioning. Reviewers 8cu7, uS6o  mention previous work such as EvoGen and Winoground-T2I.
2. Limited novelty for RL. Reviewer uS6o, YYr2 do not find Pair GRPO to be a significant contribution.
3. Soundness of evaluation protocol: Reviewer TGyE has a concern about reward hacking (InternVL2.5-26B is used as a reward model and an evaluator). Reviewer 8cu7 has a concern about “simplistic” binary evaluation.
4. Limited experimental results: Reviewer TGyE asks for the SFT baseline. Reviewer 8cu7 asks for a Flow-GRPO baseline, which performs better than the proposed approach on GenEval.
5. Qualitative results: Reviewer YYr2, uS6o are not convinced by the qualitative result of the proposed model.

**Reviewer Concerns:**

1. The authors argue they use the image-first approach instead of the prompt-first approach.
3. The authors respond by using the experiments with Qwen2.5-VL-72B as an evaluator as the main result instead. Also, the authors present the results where they break prompts into sub-questions and show there is no improved reliability.
4. The authors add the requested baselines and effectively show the benefit of RL (Pair GRPO) and the superiority to Flow-GRPO on other datasets.
5. The authors present more results and claim their proposed model generates better aesthetically.

The AC sees 1, 2, 3, and 5 as only partially resolved as they are symptoms of major issues. In particular, the work requires significant re-positioning with respect to similar work to clarify its significance. Further, the paper requires more evidence to prove the reliability and the soundness of the evaluation protocol; the paper has a limited human evaluation section.

**Reviewer Scores:**

TGyE (6+): Keep or increase.
8cu7, uS6o (4): likely to keep.
YYr2 (6): likely to keep.

---

### Decision · Program_Chairs · 2026-01-26

Reject